# On the Sample Complexity of Policy Gradient Algorithm with Occupancy Approximation for General Utility Reinforcement Learning

## Abstract

Reinforcement learning with general utilities has recently gained attention thanks to its ability to unify several problems, including imitation learning, pure exploration, and safe RL. However, prior work for solving this general problem in a unified way has only focused on the tabular setting. This is restrictive when considering larger state-action spaces because of the need to estimate occupancy measures during policy optimization. In this work, we address this issue and propose to approximate occupancy measures within a function approximation class using maximum likelihood estimation (MLE). We propose a simple policy gradient algorithm (PG-OMA) where an *actor* updates the policy parameters to maximize the general utility objective whereas a *critic* approximates the occupancy measure using MLE. We provide a statistical complexity analysis of PG-OMA showing that our occupancy measure estimation error only scales with the dimension of our function approximation class rather than the size of the state action space. Under suitable assumptions, we establish first order stationarity and global optimality performance bounds for the proposed PG-OMA algorithm for nonconcave and concave general utilities respectively. We complement our methodological and theoretical findings with promising empirical results showing the scalability potential of our approach compared to existing tabular count-based approaches.

## 1 Introduction

Reinforcement learning with general utilities (RLGU) has emerged as a general framework to unify a range of RL applications where the objective of the RL agent cannot be simply cast as a standard expected cumulative reward (Zhang et al., 2022). For instance, in imitation learning, the objective is to learn a policy by minimizing the divergence between the state-action occupancy measure induced by the policy and expert demonstrations (Ho & Ermon, 2016). In pure exploration, the goal is to learn a policy to explore the state space in a reward-free setting by maximizing the entropy of the state occupancy measure induced by the agent's policy (Hazan et al., 2019). Other examples include risk-averse and constrained RL (García & Fernández, 2015), diverse skills discovery (Eysenbach et al., 2019), and experiment design (Mutny et al., 2023).

It is well known that the standard RL objective can be written as a linear functional of the occupancy measure. To capture all the aforementioned applications, the RLGU objective is a possibly nonlinear functional of the state action occupancy measure induced by the policy (Zhang et al., 2022). Due to non-linearity, policy gradient algorithms for solving RLGU problems face the major bottleneck of occupancy measure estimation. Prior works (Hazan et al., 2019; Zhang et al., 2022) have focused on the tabular setting where the state action occupancy measure needs to be estimated for *each* state action pair using Monte Carlo estimation via sampling trajectories. However, this setting is restrictive for larger state and actions spaces where tabular methods will become intractable due to the curse of dimensionality. This scalability issue stands as an important challenge to overcome to establish RLGU as a general unified framework for which efficient algorithms exist to solve its larger state action space instances. We refer the reader to Figure 1 for an illustration of the challenge motivating our work. Our goal is to address this scalability challenge by proposing a simple algorithm for the general and flexible RLGU framework. In the standard RL setting, several approaches using function approximation have been fruitfully used to approximate action-value functions and scale

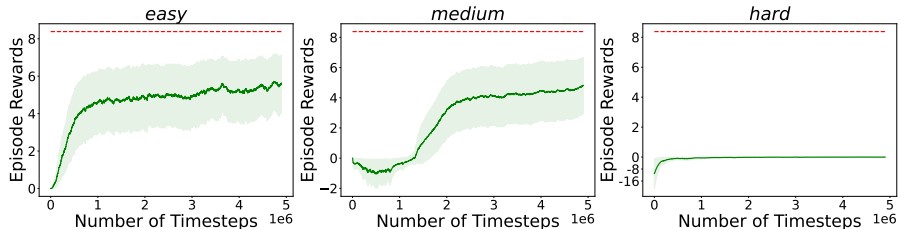

Figure 1: *A Motivating Example:* This figure shows the scalability performance of state-of-the-art count-based method of Zhang et al. (2021) in the RLGU setting for a specific application of learning from demonstration (detailed in Sec. 5). We consider three settings *easy*, *medium*, and *hard* and report the episode reward returns. The "easy" setting (**left**) has $10^2$ states, the *medium* setting (**middle**) features $10^3$ states, and the *harder* setting (**right**) comprises $10^4$ states. In the *easy* setting, the count-based method performs relatively well, as expected, since it aims to precisely estimate the occupancy measure (we employ a batch size of $B = 100$ for estimating the occupancy in each episode). However, as we transition to larger state space settings, it fails to perform due to scalability issues in estimating occupancy measures. This renders the existing general utility RL approach practically inapplicable. The red dotted line shows the oracle's performance.

to large state-action spaces. However, to the best of our knowledge, this issue remains open for RL problems with general utilities. To this end, we summarize our contributions as follows.

**Main contributions.** In this work, we propose to go beyond the tabular setting in solving RL problems with general utilities. Our contributions are summarized as follows:

- **We propose a new policy gradient algorithm, PG-OMA,** to solve RLGU where an actor performs policy parameter updates whereas a critic approximates the state-action occupancy measure via maximum likelihood estimation (MLE) within a function approximation class (cf. Sec. 3).

- **Theoretical results.** We analyze the sample complexity of our algorithm under suitable assumptions. Our analysis relies on a total variation performance bound for occupancy measure approximation via MLE which scales with the dimension of the parameters of the function approximation class rather than the state action space size. Using this result, we establish first-order stationarity and global optimality guarantees for our algorithm for nonconcave and concave general utilities respectively (cf. Sec. 4).

- **Experimental evaluations.** We conduct experiments on discrete and continuous state-action space environments for learning from demonstration tasks (cf. Sec. 5) to complement our theoretical analysis and show the scalability potential of our approach compared to existing tabular count-based approaches.

**Related Works.** The general framework of RLGU, also known as *convex RL*, has been recently introduced in the literature Hazan et al. (2019); Zhang et al. (2021); Zahavy et al. (2021); Geist et al. (2022). Hazan et al. (2019) initially focused on the particular instance of maximum entropy exploration problem and Zhang et al. (2020) proposed a variational policy gradient method to solve the RLGU problem. Zhang et al. (2021) then introduced a simpler (variance-reduced) policy gradient method to solve the (possibly nonconcave) RL problem with general utilities using a simpler policy gradient theorem (see also Kumar et al. (2022)). Later, Barakat et al. (2023) proposed an even simpler single-loop normalized policy gradient algorithm to solve RLGU. Zahavy et al. (2021) leveraged Fenchel duality to cast the convex RL problem into a saddle-point problem that can be solved using standard RL algorithms. In a line of works, Mutti et al. (2022b;a; 2023) formulated the convex RL problem in finite trials instead of infinite realizations and considered an objective which is any convex function of the empirical state distribution computed from a finite number of realizations. Ying et al. (2023a) introduced policy-based primal-dual methods for solving convex constrained CMDPs and Ying et al. (2023b) further addressed a multi-agent RL problem with general utilities. All the aforementioned works focus on the tabular setting. In particular, most of these works use a count-based Monte Carlo estimate of the occupancy measure that cannot scale to large state-action spaces. More recently, Huang et al. (2023) provided sample-efficient online/offline RL

algorithms with density features in low-rank MDPs for occupancy estimation. See appendix A for an extended related work discussion.

**Notations.** For a given finite set $\mathcal{X}$, we use the notation $|\mathcal{X}|$ for its cardinality and $\Delta(\mathcal{X})$ for the space of probability distributions over $\mathcal{X}$. We equip any Euclidean space with its standard inner product denoted by $\langle \cdot, \cdot \rangle$. The notation $\| \cdot \|$ refers to both the standard 2-norm for vectors and the spectral norm for matrices. For any vector $\lambda \in \mathbb{R}^d$ where $d$ is an integer, the notation $\lambda \geq 0$ means that all the coordinates of the vector $\lambda$ are non-negative. We interchangeably denote functions $f : \mathcal{X} \to \mathbb{R}$ over a finite set $\mathcal{X}$ as vectors $f \in \mathbb{R}^{|\mathcal{X}|}$ with components $f(x)$ with a slight abuse of notations.

## 2 PROBLEM FORMULATION

**MDP with General Utility.** Consider a discrete-time discounted Markov Decision Process (MDP) $(\mathcal{S}, \mathcal{A}, \mathcal{P}, F, \rho, \gamma)$, where $\mathcal{S}$ and $\mathcal{A}$ are finite state and action spaces respectively, $\mathcal{P} : \mathcal{S} \times \mathcal{A} \to \Delta(\mathcal{S})$ is the state transition probability kernel, $F : \mathcal{M}(\mathcal{X}) \to \mathbb{R}$ is a general utility function defined over the space of measures $\mathcal{M}(\mathcal{X})$ on the product state-action space $\mathcal{X} := \mathcal{S} \times \mathcal{A}$, $\rho$ is the initial state distribution, and $\gamma \in (0, 1)$ is the discount factor. A stationary policy $\pi : \mathcal{S} \to \Delta(\mathcal{A})$ maps each state $s \in \mathcal{S}$ to a distribution $\pi(\cdot|s)$ over the action space $\mathcal{A}$. The set of all stationary policies is denoted by $\Pi$. At each time step $t \in \mathbb{N}$ in a state $s_t \in \mathcal{S}$, the RL agent chooses an action $a_t \in \mathcal{A}$ with probability $\pi(a_t|s_t)$ and then environment transitions to a state $s_{t+1} \in \mathcal{S}$ with probability $\mathcal{P}(s_{t+1}|s_t, a_t)$. We denote by $\mathbb{P}_{\rho,\pi}$ the probability distribution of the Markov chain $(s_t, a_t)_{t \in \mathbb{N}}$ induced by the policy $\pi$ with initial state distribution $\rho$. We use the notation $\mathbb{E}_{\rho,\pi}$ (or often simply $\mathbb{E}$) for the associated expectation. We define for any policy $\pi \in \Pi$ the (normalized) state and state-action occupancy measures $d^\pi \in \mathcal{M}(\mathcal{S}), \lambda^\pi \in \mathcal{M}(\mathcal{S} \times \mathcal{A})$ respectively as:

$$d^\pi(s) := (1 - \gamma) \sum_{t=0}^{+\infty} \gamma^t \mathbb{P}_{\rho,\pi}(s_t = s); \quad \lambda^\pi(s, a) := d^\pi(s) \, \pi(a|s). \tag{1}$$

The general utility function $F$ assigns a real to each occupancy measure $\lambda^\pi$ induced by a policy $\pi \in \Pi$. We note that $\lambda^\pi$ will also be seen as a vector of the Euclidean space $\mathbb{R}^{|\mathcal{S}| \cdot |\mathcal{A}|}$. In the rest of this work, we will consider a class of policies parametrized by a vector $\theta \in \mathbb{R}^d$ for some fixed integer $d \in \mathbb{N}$. We shall denote by $\pi_\theta \in \Pi$ such a policy in this class.

**Policy optimization.** The goal of the RL agent is to find a policy $\pi_\theta$ solving the problem:

$$\max_{\theta \in \mathbb{R}^d} F(\lambda^{\pi_\theta}), \tag{2}$$

where $\lambda$ is defined in (1), $F$ is a smooth function supposed to be upper bounded and $F^\star$ is used to denote the maximum in (2). The agent has access to trajectories of finite length $H$ generated from the MDP under the initial distribution $\rho$ and the policy $\pi_\theta$. In particular, provided a time horizon $H$ and a policy $\pi_\theta$ with $\theta \in \mathbb{R}^d$, the learning agent can simulate a trajectory $\tau = (s_0, a_0, \cdots, s_{H-1}, a_{H-1})$ from the MDP when the state transition kernel $\mathcal{P}$ is unknown. This general utility problem was described, for instance, in Zhang et al. (2021) (see also Kumar et al. (2022)). Recall that the standard RL problem corresponds to the particular case where the general utility function is a linear function, i.e., $F(\lambda^{\pi_\theta}) = \langle r, \lambda^{\pi_\theta} \rangle$ for some vector $r \in \mathbb{R}^{|\mathcal{S}| \cdot |\mathcal{A}|}$, in which case we recover the expected return function as an objective:

$$V^{\pi_\theta}(r) := \mathbb{E}_{\rho,\pi_\theta} \left[ \sum_{t=0}^{+\infty} \gamma^t r(s_t, a_t) \right]. \tag{3}$$

**Examples.** We provide two motivating examples of the RLGU framework as follows.

(1) *Pure Exploration:* The problem consists in finding a policy to explore a state space in the absence of a reward signal. A natural objective is to search for a policy that maximizes the entropy of the induced distribution over the state space. In this case, we have $F(\lambda^{\pi_\theta}) = - \sum_{s \in \mathcal{S}} \mu^{\pi_\theta}(s) \log \mu^{\pi_\theta}(s)$ where for every $s \in \mathcal{S}, \mu^{\pi_\theta}(s) := (1 - \gamma) \sum_{a \in \mathcal{A}} \lambda^{\pi_\theta}(s, a)$. See Hazan et al. (2019).

(2) *Learning from Demonstrations:* The goal is to learn a policy from expert behavior trajectories or demonstrations. A formulation of such a problem consists in minimizing the Kullback-Leibler

divergence w.r.t. the expert's occupancy measure induced by an unknown policy $\pi_E$, in which case $F(\lambda^{\pi_\theta}) = \langle \lambda^{\pi_\theta}, r \rangle - c\mathrm{KL}(\lambda^{\pi_\theta} || \lambda^{\pi_E})$. See Ho & Ermon (2016); Kang et al. (2018).

**Remark 1.** *We prefer the terminology of 'RL with general utilities' to 'convex RL' since the objective may even be nonconvex in the occupancy measure in full generality. Although our focus in this work is on concave utilities in experiments, we provide first-order stationarity theoretical guarantees for the nonconcave case. While the convex RL literature exclusively focuses on the case of concave utilities, a lot of applications of interest do not fall under this umbrella and inherently involve nonconcave utilities. We provide several such examples in Appendix C.*

# 3 POLICY GRADIENT ALGORITHM WITH OCCUPANCY MEASURE APPROXIMATION (PG-OMA)

In this section, we propose a policy gradient algorithm to solve the policy optimization problem (2) with general utilities for larger state-action spaces. We start by elaborating on the challenges faced to solve such a large-scale problem. Section 3.1 mainly contains known material from the recent literature (Zhang et al., 2021), we report it here separately from the problem formulation in section 2 to motivate our algorithmic design. The rest of the section presents our algorithmic contributions.

## 3.1 POLICY GRADIENT THEOREM AND CHALLENGES FOR LARGE-SCALE RLGU

**Policy Gradient for RLGU.** Following the exposition in Zhang et al. (2021); Barakat et al. (2023), we derive the policy gradient for the general utility objective. For convenience, we use the notation $\lambda(\theta)$ for $\lambda^{\pi_\theta}$. Since the cumulative reward can be rewritten more compactly $V^{\pi_\theta}(r) = \langle \lambda^{\pi_\theta}, r \rangle$, it follows from the policy gradient theorem that:

$$[\nabla_\theta \lambda(\theta)]^T r = \nabla_\theta V^{\pi_\theta}(r) = \mathbb{E}_{\rho, \pi_\theta} \left[ \sum_{t=0}^{+\infty} \gamma^t r(s_t, a_t) \sum_{t'=0}^{t} \nabla \log \pi_\theta(a_{t'}|s_{t'}) \right], \quad (4)$$

where $\nabla_\theta \lambda(\theta)$ is the Jacobian matrix of the vector mapping $\lambda(\theta)$. Using the chain rule, we have

$$\nabla_\theta F(\lambda(\theta)) = [\nabla_\theta \lambda(\theta)]^T \nabla_\lambda F(\lambda(\theta)) = \nabla_\theta V^{\pi_\theta}(r)|_{r=\nabla_\lambda F(\lambda(\theta))}. \quad (5)$$

The classical policy gradient in the standard RL setting uses rewards which are obtained via interaction with the environment. In RLGU, there is no reward function but rather a *pseudoreward* $\nabla_\lambda F(\lambda(\theta))$ depending on the unknown occupancy measure induced by the policy.

**Stochastic Policy Gradient.** In view of performing a stochastic policy gradient algorithm, we would like to estimate the policy gradient $\nabla_\theta F(\lambda(\theta))$ in (5). We can use the standard reinforce estimator suggested by Eq. (4). Define for every reward function $r$ (which is also seen as a vector in $\mathbb{R}^{|\mathcal{S}| \times |\mathcal{A}|}$), every $\theta \in \mathbb{R}^d$ and every $H$-length trajectory $\tau$ simulated from the MDP with policy $\pi_\theta$ and initial distribution $\rho$ the (truncated) policy gradient estimate:

$$g(\tau, \theta, r) = \sum_{t=0}^{H-1} \left( \sum_{h=t}^{H-1} \gamma^h r(s_h, a_h) \right) \nabla \log \pi_\theta(a_t|s_t). \quad (6)$$

Given (5), we also need to estimate the state-action occupancy measure $\lambda(\theta)$ (when $F$ is nonlinear)[1]. Prior work has exclusively focused on the tabular setting using a Monte-Carlo estimate of this occupancy measure $\lambda^{\pi_\theta} = \lambda(\theta)$ (see (1)) truncated at the horizon $H$ by $\lambda(\tau) = \sum_{h=0}^{H-1} \gamma^h \delta_{s_h, a_h}$ where for every $(s, a) \in \mathcal{S} \times \mathcal{A}$, $\delta_{s,a} \in \mathbb{R}^{|\mathcal{S}| \times |\mathcal{A}|}$ is a vector of the canonical basis of $\mathbb{R}^{|\mathcal{S}| \times |\mathcal{A}|}$, i.e., the vector whose only non-zero entry is the $(s, a)$-th entry which is equal to $1$, and $\tau = \{(s_h, a_h)\}_{0 \le h \le H-1}$ is a trajectory of length $H$ generated by the MDP controlled by the policy $\pi_\theta$.

**Challenges for Large-scale RLGU.** One of the main challenges in solving the general utility problem (2) via a policy gradient algorithm based on (5) is to estimate the unknown state-action occupancy measure $\lambda(\theta)$ in large scale settings involving huge state and action spaces. This problem is

---

[1]In the cumulative reward setting, the utility $F$ is linear w.r.t. $\lambda$ and $\nabla_\lambda F(\lambda(\theta))$ is independent of $\lambda(\theta)$.

arguably more delicate than that of estimating action-value functions in cumulative expected reward RL problems. First, while action-value functions satisfy a *forward* Bellman equation, occupancy measures satisfy a *backward* Bellman flow equation. This fundamental difference makes it hard to design stochastic algorithms minimizing mean-square Bellman errors as it is customary in algorithms using function approximation to solve standard RL problems (see end of appendix A for further explanations). Second and foremost, while prior work has used Monte Carlo estimates for this quantity, such count-based estimates are not tractable beyond small tabular settings. Indeed, for very large state-action spaces, it is not tractable to compute and store a table of count-based estimates of the true occupancy measure containing all the values for all the state-action pairs. In the next section, we propose an approach to tackle this issue.

**Remark 2.** *(**Extension to continuous state-action spaces**) Our algorithm can be used in the continuous (compact) state-action space setting since it only relies on using policy gradients and MLE which are both scalable. We stick to the discrete state action space notation for ease of exposition to avoid the technical measure theoretical formalism to address the continuous setting in full mathematical rigor.*

### 3.2 Occupancy Measure Estimation

In this section, we address the challenge of occupancy measure estimation in large state action spaces. Given a policy $\pi_\theta$, our goal is to estimate the unknown occupancy measure $d^{\pi_\theta}$ induced by this policy using state samples obtained from executing the policy. Since the normalized occupancy measure is a probability distribution, we propose to perform maximum likelihood estimation. Before presenting this procedure, we elaborate on the motivation behind approximating the occupancy measure by a parametrized distribution in a given function class of neural networks for example.

**Motivation.** Besides the practical motivation of using distribution approximation to scale to larger state-action space settings, we provide some theoretical motivation. Recall that action-value functions are linear in the feature map for linear (or low-rank) MDPs for solving standard cumulative sum RL problems (see Proposition 2.3 in Jin et al. (2019)). Similarly, it turns out that state-occupancy measures are linear (or affine in the discounted setting) in density features in low-rank MDPs. We refer the reader to Appendix B for a proof of this statement (see also Lemma 16, 17 in Huang et al. (2023)). Therefore, in this case, it is natural to approximate occupancy measures via linear function approximation using some density features. More generally, for an arbitrary MDP, we propose to approximate the (normalized) state occupancy measure $d^{\pi_\theta}$ induced by a policy $\pi_\theta$ directly by a probability distribution in a certain parametric class of probability distributions:

$$\Lambda := \{p_\omega \in \Delta(\mathcal{S}) \,|\, \omega \in \Omega \subseteq \mathbb{R}^m\}\,, \tag{7}$$

where for instance $m \ll |\mathcal{S}|$. An example of such a parametrization for a given $\omega \in \mathbb{R}^m$ is the softmax $\sigma_\omega$ defined over the state space by $\sigma_\omega(s) := \exp(\psi_\omega(s))/Z(\omega)$, where $Z(\omega) := \sum_{s' \in \mathcal{S}} \exp(\psi_\omega(s'))$ and where $\psi_\omega : \mathcal{S} \to \mathbb{R}$ is a given mapping which can be a neural network in practice. For continuous state spaces, practitioners can consider for instance Gaussian mixture models with means and covariance matrices encoded by trainable neural networks.

**Maximum Likelihood Estimation (MLE).** For simplicity, we suppose we have access to i.i.d. state samples following the distribution $d^{\pi_\theta}$ throughout our exposition. We refer the reader to Appendix D.1 for a discussion about how to sample such states. Given the parametric distribution class $\Lambda$ defined in (7) and a data set $\mathcal{D} := \{s_i\}_{i=1,\cdots,n} \in \mathcal{S}^n$ of $n$ i.i.d. state samples following the distribution $d^{\pi_\theta}$ induced by the current policy $\pi_\theta$, we construct the standard MLE

$$\hat{d}^{\pi_\theta} := p_{\omega^*}\,, \qquad \omega^* \in \arg\max_{\omega \in \Omega} \frac{1}{n} \sum_{i=1}^n \log p_\omega(s_i)\,. \tag{8}$$

An estimator of the state-action occupancy measure $\lambda^{\pi_\theta}$ is then given by $\hat{\lambda}^{\pi_\theta}(s, a) = \hat{d}^{\pi_\theta}(s)\,\pi_\theta(a|s)$ for any $s \in \mathcal{A}, a \in \mathcal{A}$ (see (1)). Using MLE is important for our scalability goal. Barakat et al. (2023) recently proposed a different procedure based on mean square error estimation. Please see appendix A for a detailed comparison with this work highlighting the merits of our approach. In practice, a neural network learns the parameters of a chosen parametrized distribution class for approximating the true occupancy measure by maximizing the log-likelihood loss (8) over the samples generated (see appendix D.1 for sampling).

### 3.3 PROPOSED ALGORITHM

Based on our discussion in sections 3.1 and 3.2, we propose a simple stochastic policy gradient algorithm which consists of two main steps:

  (i) Compute an approximation of the unknown state-action occupancy measure $\lambda^{\pi_\theta} \in \mathbb{R}^{|\mathcal{S}| \times |\mathcal{A}|}$ for a fixed parameter $\theta \in \mathbb{R}^d$ with MLE using collected state samples (see (8));

  (ii) Perform stochastic policy gradient ascent using the stochastic policy gradient defined in (6) using the estimated occupancy measure computed in the first step.

The resulting algorithm is Algorithm 1 which is model-free as we do not estimate the transition kernel.

---

**Algorithm 1** PG for RLGU with Occupancy Measure Approximation (PG-OMA)

---

1: **Input:** $\theta_0 \in \mathbb{R}^d, T, N \geq 1, \alpha > 0, H$ .
2: **for** $t = 0, \ldots, T - 1$ **do**
    //Occupancy approximation for pseudo-reward learning
3:    Compute the MLE estimator $\hat{\lambda}_t = \hat{d}^{\pi_{\theta_t}} \cdot \pi_{\theta_t}$ using policy $\pi_{\theta_t}$ (see (8)).
4:    $\hat{r}_t = \nabla_\lambda F(\hat{\lambda}_t)$
    //Policy parameter update
5:    Sample a batch of $N$ independent trajectories $(\tau_t^{(i)})_{1 \leq i \leq N}$ of length $H$ using $\pi_{\theta_t}$ .
6:    $\theta_{t+1} = \theta_t + \frac{\alpha}{N} \sum_{i=1}^N g(\tau_t^{(i)}, \theta_t, \hat{r}_t)$ (see (6))
7: **end for**
8: **Return:** $\theta_T$

---

**Remark 3.** *When running Algorithm 1, note that the vector $\hat{\lambda}_t \in \mathbb{R}^{|\mathcal{S}| \times |\mathcal{A}|}$ (and hence the vector $r_t$) is not computed for all state-action pairs. Indeed, at each iteration, one does only need to compute $(r_t(s_h^{(t)}, a_h^{(t)}))_{0 \leq h \leq H-1}$ where $\tau_t = (s_h^{(t)}, a_h^{(t)})_{0 \leq h \leq H-1}$ to obtain the stochastic policy gradient $g(\tau_t, \theta_t, r_{t-1})$ as defined in (6).*

Our occupancy measure estimation step can be seen as a critic for pseudo-reward learning. Notice though that this critic is not approximating a value function like in standard RL but rather the occupancy measure which is a distribution.

## 4 CONVERGENCE AND SAMPLE COMPLEXITY ANALYSIS

### 4.1 STATISTICAL COMPLEXITY OF OCCUPANCY MEASURE ESTIMATION

In this section, we suppose we are given a data set of i.i.d. state-action pair samples following the (normalized) occupancy measure $\lambda^\pi$ induced by a fixed given policy $\pi$ . As previously explained, we approximate $\lambda^\pi$ by a function (or parametrized density) in the function class $\Lambda$ defined in (7). We make the following assumption to control the complexity of our function approximation class.

**Assumption 1** (Function approximation class regularity). *The following holds true:*

  *(i) (parameter compactness) The set $\Omega$ is compact, we denote by $B_\omega := \max_{\omega \in \Omega} \|\omega\|_\infty$;*

  *(ii) (realizability) The (normalized) occupancy measure to be estimated satisfies: $\lambda^\pi \in \Lambda$;*

  *(iii) (Lipschitzness) $\forall \omega, \bar{\omega} \in \Omega, \forall x \in \mathcal{X}, \exists L(x) \in \mathbb{R}$ s.t. $|p_\omega(x) - p_{\bar{\omega}}(x)| \leq L(x)\|\bar{\omega} - \omega\|_\infty$ with $B_L := \int_\mathcal{X} L(x)dx < +\infty$ .*

Assumption 1 is satisfied for instance for the class of generalized linear models, i.e. $\Lambda := \{p_\omega(x) = g(\omega^T \phi(x)), \forall x \in \mathcal{X} : p_\omega \in \Delta(\mathcal{X}), \omega \in \Omega\}$ where $g : \mathbb{R} \to [0, 1]$ is an increasing Lipschitz continuous function and $\phi : \mathcal{X} \to \mathbb{R}^d$ is a given feature map s.t. $\int \|\phi(x)\|_1 dx \leq B_L$ for some $B_L > 0$. Notice that features can be normalized appropriately to satisfy the assumption. A similar assumption has been made in the case of linear MDPs in (Huang et al., 2023, Assumption 1). The realizability assumption can be relaxed at the price of incurring a misspecification error.

We now state our sample complexity result for occupancy measure estimation via MLE in view of our PG sample complexity analysis. This result relies on arguments developed in the statistics literature Van de Geer (2000); Zhang (2006). These techniques were adapted to the RL setting for low-rank MDPs in e.g. Agarwal et al. (2020). Our proof builds on Huang et al. (2023) which we slightly adapt for our purpose (see Appendix D.2).

**Proposition 1.** *Let Assumption 1 hold true. Then for any $\delta > 0$, the MLE $\hat{\lambda}^{\pi_\theta}$ defined using (8) satisfies with probability at least $1 - \delta$,*
$$\|\hat{\lambda}^{\pi_\theta} - \lambda^{\pi_\theta}\|_1 \leq 6\sqrt{\frac{12\, m \log\left(\frac{2\lceil B_\omega B_L n\rceil}{\delta}\right)}{n}}\,.$$

The above result translates into a sample complexity of $\tilde{\mathcal{O}}(m\,\varepsilon^{-2})$ to guarantee an $\varepsilon$-approximation of the true occupancy measure (in the $l_1$-norm distance) using samples. We highlight that our sample complexity only depends on the dimension $m$ of the parameter space and does not scale with the size of the state-action space. Hence the MLE procedure we use is the key ingredient to scale our algorithm to large state-action spaces. To the best of our knowledge, existing algorithms for solving the RLGU problem (with nonlinear utility functions) are limited to the restrictive tabular setting.

## 4.2 Guarantees for Policy Gradient with Occupancy Measure Approximation

In this section, we establish sample complexity guarantees for Algorithm 1. We start by introducing the assumptions required for our results and discuss their relevance.

**Assumption 2** (Policy parametrization). *The following holds for every $(s, a) \in \mathcal{S} \times \mathcal{A}$. For every $\theta \in \mathbb{R}^d$, $\pi_\theta(a|s) > 0$. Moreover, the function $\theta \mapsto \pi_\theta(a|s)$ is continuously differentiable and the score function $\theta \mapsto \nabla \log \pi_\theta(a|s)$ is bounded by some positive constant $B$.*

This standard assumption is satisfied for instance by the common softmax policy parametrization defined for every $\theta \in \mathbb{R}^d, (s, a) \in \mathcal{S} \times \mathcal{A}$ by $\pi_\theta(a|s) = \frac{\exp(\psi(s,a;\theta))}{\sum_{a' \in \mathcal{A}} \exp(\psi(s,a';\theta))}$, where $\psi : \mathcal{S} \times \mathcal{A} \times \mathbb{R}^d \to \mathbb{R}$ is a smooth function such that the map $\psi(s, a; \cdot)$ is twice continuously differentiable for every $(s, a) \in \mathcal{S} \times \mathcal{A}$ and for which there exist $l_\psi, L_\psi > 0$ s.t. (i) $\max_{s \in \mathcal{S}, a \in \mathcal{A}} \sup_\theta \|\nabla \psi(s, a; \theta)\| \leq l_\psi$ and (ii) $\max_{s \in \mathcal{S}, a \in \mathcal{A}} \sup_\theta \|\nabla^2 \psi(s, a; \theta)\| \leq L_\psi$.

**Assumption 3** (General utility smoothness). *There exist constants $l_\lambda, L_\lambda > 0$ s.t. for all $\lambda_1, \lambda_2 \in \Lambda$, $\|\nabla_\lambda F(\lambda_1)\|_2 \leq l_\lambda$ and $\|\nabla_\lambda F(\lambda_1) - \nabla_\lambda F(\lambda_2)\|_2 \leq L_\lambda \|\lambda_1 - \lambda_2\|_2$.*

Under Assumptions 2 and 3, the function $\theta \mapsto F(\lambda^{\pi_\theta})$ is $L_\theta$-smooth (see Lemma 3 for the expression). Using this property, the next result shows that our algorithm enjoys a first-order stationary guarantee in terms of the non-convex general utility objective.

**Theorem 1.** *(Nonconcave general utility) Let Assumptions 2, 3 hold. Then the iterates generated by Algorithm 1 with step sizes $\alpha_t \leq 1/(2L_\theta)$ and $T \geq 1$ iterations satisfy:*
$$\mathbb{E}[\|\nabla_\theta F(\lambda^{\pi_{\theta_\tau}})\|^2] \leq \frac{16(F^\star - \mathbb{E}[F(\lambda^{\pi_{\theta_1}})])}{\alpha T} + \frac{C_1}{N} + C_2 \mathbb{E}[\|\hat{\lambda}_\tau - \lambda^{\pi_{\theta_\tau}}\|_2^2]\,, \qquad (9)$$
*where $\tau$ is a uniform random variable over $\{1, \cdots, T\}$ and expectation is w.r.t. all randomness (in $(\theta_t)$ and $\tau$).*

The above upper bound shows a decomposition of the first order stationarity error into three terms: the first two are the typical errors incurred by PG methods whereas the third one is due to occupancy measure approximation. In particular, choosing the number of iterations $T$, the batch size $N$ (of sampled trajectories) appropriately and the number $n$ of samples used in MLE for occupancy measure approximation, we obtain the following sample complexity result.

**Corollary 1.** *Let Assumptions 1, 2, 3 hold. Setting the number of iterations to $T = \mathcal{O}(\epsilon^{-1})$, the batch size for PG to $N = \mathcal{O}(\epsilon^{-1})$ and the number of samples for occupancy measure MLE to $n = \mathcal{O}(m\epsilon^{-1})$ for some precision $\epsilon > 0$ in Theorem 1, it holds that $\mathbb{E}[\|\nabla_\theta F(\lambda^{\pi_{\theta_\tau}})\|^2] \leq \epsilon$. The total sample complexity is then $T(N + n) = \mathcal{O}(m\epsilon^{-2})$.*

In several applications in RLGU, the utility function $F$ is concave w.r.t. its occupancy measure variable. We now turn to proving global performance bounds under this particular setting.

**Assumption 4** (Concavity). *The utility function $F : \Lambda \to \mathbb{R}$ is concave.*

Notice that the general utility objective is in general nonconcave w.r.t. the policy parameter $\theta$. Despite this non-concavity, we can exploit the so-called *hidden convexity* (concavity in our setting) of the problem Zhang et al. (2021). We require an additional regularity assumption on the policy parametrization which has been previously made in Zhang et al. (2021); Ying et al. (2023a); Barakat et al. (2023). While this assumption holds for a tabular policy parametrization, it is delicate to relax it further, see e.g. (Barakat et al., 2023, Appendix C) for a discussion.

**Assumption 5** (Policy overparametrization). *For the softmax policy parametrization defined above, the following three requirements hold: (i) For any $\theta \in \mathbb{R}^d$, there exist relative neighborhoods $\mathcal{U}_\theta \subset \mathbb{R}^d$ and $\mathcal{V}_{\lambda(\theta)} \subset \Lambda$ respectively containing $\theta$ and $\lambda(\theta)$ s.t. the restriction $\lambda|_{\mathcal{U}_\theta}$ forms a bijection between $\mathcal{U}_\theta$ and $\mathcal{V}_{\lambda(\theta)}$; (ii) There exists $l > 0$ s.t. for every $\theta \in \mathbb{R}^d$, the inverse $(\lambda|_{\mathcal{U}_\theta})^{-1}$ is $l$-Lipschitz continuous; (iii) There exists $\bar{\eta} > 0$ s.t. for every positive real $\eta \leq \bar{\eta}$, $(1 - \eta)\lambda(\theta) + \eta\lambda(\theta^*) \in \mathcal{V}_{\lambda(\theta)}$ where $\pi_{\theta^*}$ is the optimal policy.*

The following result makes use of the concavity of the utility function $F$ to obtain a global optimality guarantee for the iterates of our algorithm under the assumption that the occupancy measures induced by the policies encountered during the run of the algorithm are uniformly well-approximated.

**Theorem 2.** *(Concave general utility) Let Assumptions 2 to 5 hold. Assume further that there exists $\epsilon_{MLE} > 0$ s.t. $\mathbb{E}[\|\hat{\lambda}_t - \lambda(\theta_t)\|_2^2] \leq \epsilon_{MLE}$ uniformly over $T \geq 1$ iterations of Algorithm 1 with step sizes $\alpha_t \leq 1/(2L_\theta)$. Then the iterate output $\theta_T$ of Algorithm 1 satisfies for any $\eta < \bar{\eta}$,*

$$\mathbb{E}[F^\star - F(\lambda(\theta_T))] \leq (1 - \eta)^T \delta_0 + C_3 \frac{\eta}{\alpha} + C_4 \frac{\alpha}{\eta} \left( \frac{1}{N} + \epsilon_{MLE} \right), \tag{10}$$

*for some positive constants $C_3, C_4$ explicit in Appendix D.4, (48) and $\delta_0 := \mathbb{E}[F^\star - F(\lambda(\theta_0))]$.*

Using the above result, we derive the following sample complexity guarantee by specifying the step size and number of iterations of our algorithm as well as large enough batch size and number of samples for MLE using Proposition 1.

**Corollary 2.** *Let Assumptions 1 to 5 hold. For any given precision $\epsilon > 0$, set $T = \frac{1}{\eta} \log(\frac{\delta_0}{\epsilon}), \alpha = \mathcal{O}(\epsilon), \eta = \mathcal{O}(\epsilon^2), N = \mathcal{O}(\epsilon^{-2})$ and $n = \mathcal{O}(m\epsilon^{-2})$, then the total sample complexity to obtain $\mathbb{E}[F^\star - F(\lambda(\theta_t))] \leq \epsilon$ is given by $T(N + n) = \mathcal{O}(m\epsilon^{-4})$.*

## 5 PROOF OF CONCEPT EXPERIMENTS

In this section, we investigate the capability of the proposed PG-OMA in terms of scaling with respect to the dimensionality of the state space when solving RLGU problems. In this work, we perform initial proof of concept experiments on simulation environments such as MPE (Multi-Agent Particle Environment (Lowe et al., 2017)) and SMAC (StarCraft Multi-Agent Challenge (Samvelyan et al., 2019)). We provide additional details about the experiments in Appendix E. We consider the problem of learning from demonstrations as defined in Example 2 in Sec. 2 and show results in discrete and continuous state space settings. Before presenting our experimental results, we want to emphasize that our experiments serve as evidence of the potential of the proposed approach in addressing scalability challenges in RLGU. We do not claim to surpass the state-of-the-art performance in solving specific tasks (of learning from demonstration) within the MPE and SMAC environments. In contrast to prior work which mostly designed tailored algorithms for specific single tasks, note that our algorithm can be used for any RLGU problem.

(1) *Discrete Spaces.* To further demonstrate the effectiveness of our proposed approach, we conducted experiments in a $10 \times 10$ gridworld environment with varying numbers of agents tasked with reaching distinct goal positions (refer to Figure 4 in the Appendix for a detailed gridworld description). It is important to note that as the number of agents in the environment increases, the

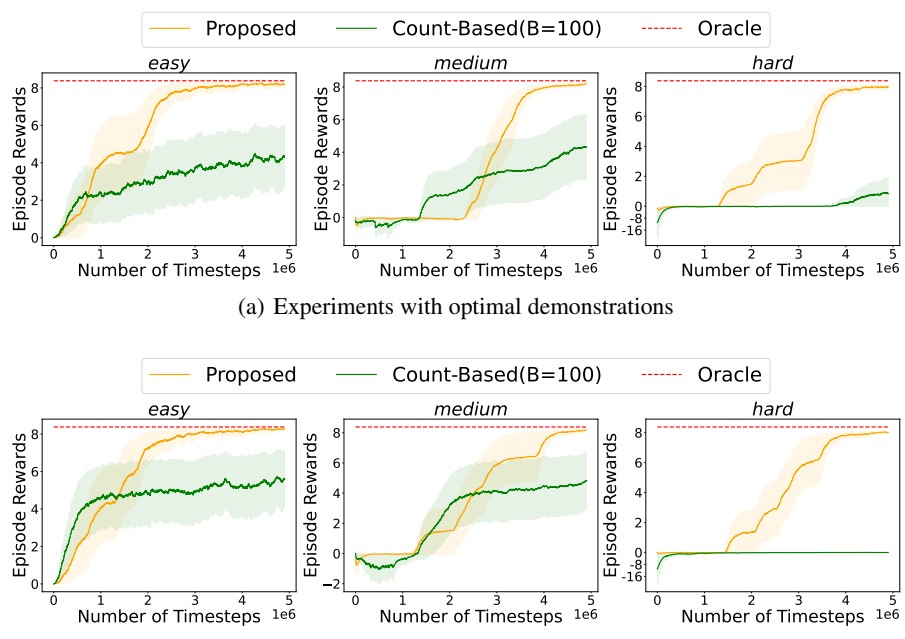

(a) Experiments with optimal demonstrations

(b) Experiments with suboptimal demonstrations

Figure 2: (a) This figure compares the convergence of our proposed approach across three distinct settings: *easy*, *medium*, and *hard*. The "easy" setting has $10^2$ states, the *medium* setting features $10^3$ states, and the *harder* setting comprises $10^4$ states. In the *easy* setting, the count-based method performs relatively well, as expected, since it aims to precisely estimate the occupancy measure (we employ a batch size of $B = 100$ for estimating the occupancy in each episode). However, as we transition to larger state space settings, our proposed method outperforms the count-based approach significantly. (b) We conducted tests with suboptimal demonstrations and show that our proposed algorithm remains effective. The shaded area is a tolerance interval (with mean and standard deviation) built from running the experiment with 5 different seeds.

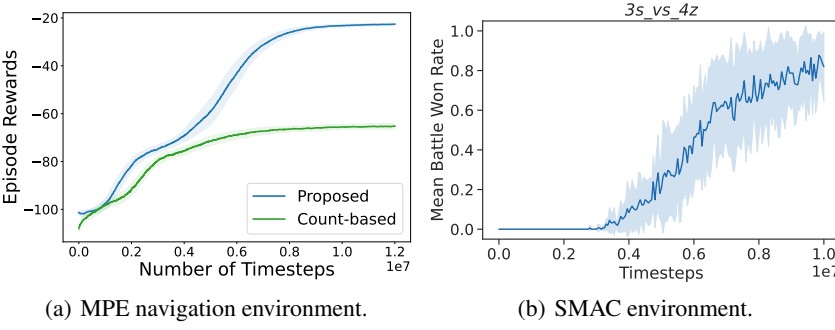

(a) MPE navigation environment.

(b) SMAC environment.

Figure 3: This figure shows the effectiveness of our proposed approach in continuous state space environments, such as MPE and SMAC environments. For MPE, we plot the performance of the base method via discretization of the state space, which clearly results in suboptimal results. We only report the results of our proposed approach for SMAC as the count-based baseline was intractable.

joint state and action space grows exponentially. Additionally, we consider a sparse reward setting, where agents receive a non-zero reward only when they successfully reach their respective goals; otherwise, the reward remains zero. This sparse reward setup makes the problem hard, requiring the incorporation of demonstrations from experts to guide learning, aligning with the RLGU problem outlined in Section 2. Figure 2 summarizes the effectiveness of the proposed approach as compared to the count-based estimation method.

(2) *Continuous Spaces.* We also conducted experiments to demonstrate the effectiveness of the proposed approach in continuous spaces on (a) the cooperative navigation task from the multi-agent particle environment (MPE) and (b) the SMAC environment based on the StarCraft II game, we consider 3sv4z, which features 3 Stalkers (allies) versus 4 Zealots (enemies). For comparison, we discretize the state space to perform the count-based estimation method as a baseline. Figure 3 presents the training curves of both methods.

## 6 CONCLUSION

In this paper, we proposed a simple policy gradient algorithm for RLGU to address the fundamental challenge of scaling to larger state-action spaces beyond the tabular setting. Our approach hinges on using MLE for approximating occupancy measures to construct a stochastic policy gradient. We proved that our MLE procedure enjoys a sample complexity which only scales with the dimension of the parameters in our function approximation class rather than the size of the state-action space which might even be continuous. Under suitable assumptions, we also provided convergence guarantees for our algorithm to first-order stationarity and global optimality respectively. We hope this work will stimulate further research in view of designing efficient and scalable algorithms for solving real-world problems.

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

CONTENTS

## A  EXTENDED RELATED WORK DISCUSSION

Table 1: Comparison to closest related works about RLGU.

| Reference | First-order stationarity rate[1] | Global optimality rate[2] | Beyond tabular[3] | No state space size dependence[4] |
|---|---|---|---|---|
| Hazan et al. (2019) | ✘ | $\tilde{\mathcal{O}}(\epsilon^{-3})^{\&}$ | ✘ | ✘ |
| Zhang et al. (2020) | $\tilde{\mathcal{O}}(\epsilon^{-2})^{*}$ | $\tilde{\mathcal{O}}(\epsilon^{-1})^{*}$ | ✘ | ✘ |
| Zhang et al. (2021) | $\tilde{\mathcal{O}}(\epsilon^{-3})^{\#}$ | $\tilde{\mathcal{O}}(\epsilon^{-2})^{\#}$ | ✘ | ✘ |
| Zahavy et al. (2021) | ✘ | $\tilde{\mathcal{O}}(\epsilon^{-3})^{\&}$ | ✘ | ✘ |
| Barakat et al. (2023) (sec. 4) | $\tilde{\mathcal{O}}(\epsilon^{-3})^{\#}$ | $\tilde{\mathcal{O}}(\epsilon^{-2})^{\#}$ | ✘ | ✘ |
| Barakat et al. (2023) (sec. 5) | $\tilde{\mathcal{O}}(\epsilon^{-4})$ | ✘ | ✓ | ✘ |
| Mutti et al. (2023)[+] | ✘ | $\tilde{\mathcal{O}}(\epsilon^{-2})^{\&}$ | ✓ | ✘ |
| **This paper** | $\tilde{\mathcal{O}}(m\epsilon^{-4})^{\S}$ | $\tilde{\mathcal{O}}(m\epsilon^{-4})^{\S}$ | ✓ | ✓ |

$\tilde{\mathcal{O}}$ hides logarithmic factors in the accuracy $\epsilon$, mainly due to the horizon length in the infinite horizon discounted reward setting.

[1] refers to the number of samples (or number of iterations in the deterministic case when specified) to achieve a given first-order stationarity $\epsilon$, i.e. $\mathbb{E}[\|\nabla_\theta F(\lambda(\bar{\theta}_T))\|] \leq \epsilon$ where $\bar{\theta}_T$ is sampled uniformly at random from the iterates of the algorithm $\{\theta_1, \cdots, \theta_T\}$ until timestep $T$.

[2] refers to the number of samples (or number of iterations in the deterministic case when specified) to achieve global optimality under convexity of the general utility function $F$ w.r.t. its occupancy measure variable, i.e. $F^* - F(\lambda(\theta_T)) \leq \epsilon$ where $F^*$ is the maximum utility achieved for an optimal policy and $\theta_T$ is the last iterate of the algorithm generated after $T$ steps.

[3] means that the large scale state action space is discussed and addressed, i.e., the work is not restricted to the tabular setting in which occupancy measures are estimated using a simple Monte Carlo (count-based) estimator for each state $s \in \mathcal{S}$. For a more extended discussion regarding this point and comparison to prior work, please see the rest of this section below.

[4] means that the performance bounds provided for first-order stationarity or global optimality do not depend on the state space size.

[&] These results do not hold for the last iterate like for the other results but rather for a mixture of policies in (Hazan et al., 2019, Theorem 4.4), an averaged occupancy measure over the iterates in (Zahavy et al., 2021, Lemma 2) and an average regret guarantee leading to a statistical (rather than computational) complexity in (Mutti et al., 2023, Theorem 5).

[*] This is for the deterministic setting only, i.e. only reporting the number of iterations required. The rate is further improved to be linear under strong convexity of the general utility function. Other results provided report sample complexities.

[#] These results make use of variance reduction in the tabular setting to obtain improved sample complexities compared to vanilla PG algorithms.

[+] This result considers a different (single trial) problem formulation compared to ours (and other works in the literature), see detailed discussion below for a comparison.

[§] $m$ refers to the dimension of the function approximation class parameter for occupancy measure approximation, see eq. (7) and section 4. It should be noted here that we suppose access to a maximizer of the log-likelihood (8) (which requires some computational complexity that we do not discuss here), this is common in sample complexity analysis. Note also that all the other results suffer from a dependence on the size of the state space (explicit or hidden in the statements).

**Comparison to Barakat et al. (2023).** The work of Barakat et al. (2023) is mostly focused on the tabular setting (secs. 1 to 4). Section 5 therein is the only relevant section to our work which focuses on the large state action space setting. We list here several fundamental differences with our work and crucial improvements in terms of scalability:

(a) **MSE vs MLE.** The aforementioned work we compare to here uses a mean squared error estimator (MSE) whereas we use a maximum likelihood estimator (MLE), this difference turns out to be crucial for scalability. This is because mean square error estimation for occupancy measure estimation fails to scale to large state action spaces. To see this, consider an even simpler setting: suppose we have an unknown distribution $p^\star$ over a space $\mathcal{X}$ and i.i.d. samples $X_i \sim p^\star$ with $i = 1, \cdots, n$. MLE provides a TV bound $\|p - p^\star\|_1 \leq \epsilon$ where the accuracy $\epsilon$ is some $|\mathcal{X}|$-independent quantity that only depends on the sample size and complexity of the hypothesis class. In stark contrast, mean square regression would lead to

$\mathbb{E}_{x \sim p^\star}[(p(x) - p^\star(x))^2] \le \epsilon$. By the Cauchy-Schwartz inequality (which is tight if the error $p(x) - p^\star(x)$ is relatively uniform over the space), we obtain $\mathbb{E}_{x \sim p^\star}[|p(x) - p^\star(x)|] \le \sqrt{\epsilon}$. While this bound is close to the TV error bound above, it has an extra $p^\star(x)$ which implies an extra $|\mathcal{X}|$ dependence compared to the MLE approach if $p^\star$ is close to uniform. This is fundamentally not scalable. Note that MLE works even for densities over continuous spaces as it is already extensively used in the statistics literature. Please see also below (in the same section) for an extended discussion regarding MLE vs MSE;

(b) **Dependence on the state space size.** Their results do not make the dependence on the state space explicit and do not show an (exclusive) dependence on the dimension $d$ of the state action feature map. It is required in their Theorem 5.4 that $\rho(s) \ge \rho_{\min}$. Notice that if $\rho$ covers the whole state space like in the uniform distribution case, then $1/\rho_{\min}$ scales as the state space size. The dependence on this quantity is not made explicit in Theorem 5.4. After a close investigation of their proof, one can spot the dependence on $1/\rho_{\min}$ (which scales with $S$) in their constants (see e.g. in the constant $\tilde{C}_2$ in eq. (130) p. 41 in the detailed version of the theorem, see also eq. (139) p. 42 and eq. (143) p. 43 for more details).

(c) **Global convergence.** In contrast to our work (see our theorem 2 and corollary 2), they only provide a first-order stationarity guarantee and they do not provide global convergence guarantees;

(d) **Technical analysis.** From the technical viewpoint, our occupancy measure MLE estimation procedure combined with our PG algorithm requires a different theoretical analysis even for our first order stationarity guarantee. Please see appendix D below;

(e) **Experiments.** They do not provide any simulations testing their algorithm in section 5 for large state action spaces, Fig. 1 therein is only for the tabular setting.

**More about MSE vs MLE.** It is known that MSE is equivalent to MLE when the errors in a linear regression problem follow a normal distribution. However, as first preliminary comments regarding the comparison to the approach in Barakat et al. (2023), we additionally note that: (a) they only discuss the finite state action space setting for which this connection to MLE is not relevant and (b) there is no discussion nor any assumption about normality of the errors or any extension to the continuous state action space setting, we also observe that the occupancy measure values are bounded between 0 and $1/(1 - \gamma)$ (or 0 and 1 for the normalized occupancies) which is a finite support that cannot be the support of a Gaussian distribution.

Beyond these first comments, let us now elaborate in more details on their approach and its potential regarding scalability to provide further clarifications. Our goal is to learn the (normalized) state occupancy measure $d^{\pi_\theta}$ induced by a given policy $\pi_\theta$ which is a probability distribution. In the discrete setting, this boils down to estimate $d^{\pi_\theta}(s)$ for every $s \in \mathcal{S}$. Note first that this quantity can be extremely small for very large state space settings which are the focus of our work, making the probabilities hard to model especially when using a regression approach.

The approach adopted in Barakat et al. (2023) consists in seeing this estimation problem as a regression problem. In more details, since the whole distribution needs to be estimated, they propose to consider an expected mean square error over the state space (rather than solving $|\mathcal{S}|$ regression problems - one for each $\lambda^{\pi_\theta}(s)$ - which is not affordable given the scalability objective). Hence the mean square loss they define is an expected error over a state distribution $\rho$ to obtain an aggregated objective. This is less usual and specific to our occupancy measure estimation problem (this aggregation is not the mean over observations). This introduces a scalability issue as we recall that we would like to estimate $d^{\pi_\theta}(s)$ for every $s \in \mathcal{S}$, so the aggregated MSE objective considered there (see eq. (11) p. 7 therein) introduces a discrepancy w.r.t. the initial objective of estimating the whole distribution.

We do not exclude that a mean square error approach under suitable statistical model assumptions might address the occupancy measure estimation problem in a scalable way for large state action spaces for the continuous setting. However, this is not addressed in Barakat et al. (2023), their regression approach needs to be amended to address issues we mentioned above to be applicable and relevant to occupancy measure estimation and we are not sure that can be even achieved to tackle the problem for both discrete and continuous settings as we do.

**Illustrative example for the limitations of MSE vs MLE for probability distribution estimation.** We provide a simple illustrative example. Consider a simple case where the distribution $p^*(x)$ to be estimated is uniform ($p^*(x) = 1/|\mathcal{X}|$ where $|\mathcal{X}|$ is the size of the state space). The estimated distribution $p(x) = 2/|\mathcal{X}|$ on one half of the space and 0 on the other-half i.e this distribution is non-uniform, assigning a higher probability to events in one part of the space and zero probability to events in the other part. The expected loss thus incurred in this scenario using regression (namely $\sum_{x \in \mathcal{X}} p^*(x)(p(x) - p^*(x))^2$) scales as $O(1/|\mathcal{X}|^2)$ after a simple computation. This means that with large cardinality of the space, it becomes impossible to detect the difference between the two models even with infinite data when doing regression, whereas MLE does not suffer from this issue.

The primary difference between regression and MLE is that MLE results in a useful TV error bound (see Zhang (2006) and (Huang et al., 2023, Lemma 12) which we make use of in our analysis) i.e $\|p - p^*\|_1 \leq \epsilon$, where $\epsilon$ is independent of the cardinality of the space $|\mathcal{X}|$ and depends only on the sample size and complexity of the hypothesis class. In contrast, in the case of regression (MSE) where the expected loss is optimized, we get

$$\mathbb{E}_{x \sim p^*} \|p - p^*\|^2 \leq \epsilon, \mathbb{E}_{x \sim p^*} \|p - p^*\| \leq \sqrt{\epsilon}, \tag{11}$$

where the second inequality stems from an application of the Cauchy-Schwartz inequality. Note that we can write the left-hand side of the above last inequality as $\sum_{x \in \mathcal{X}} |p(x) - p^*(x)| \cdot p^*(x) \leq \epsilon$, which would eventually lead to the total variation norm upper-bounded by $\sqrt{\epsilon} \times |\mathcal{X}|$, assuming $p^*(x) = 1/|\mathcal{X}|$ to be uniform for illustration, thus incurring a large error while estimating the distribution.

**Comparison to (Mutti et al., 2023, Theorem 5, section 3).** We enumerate the differences between our results and settings in the following:

1. **Problem formulation.** As mentioned in the short related work section in the main part, Mutti et al. (2023) consider a finite trial version of the convex RL problem which has its own merits (for settings where the objective itself only cares about the performance on the finite number of realizations the agent can have access to instead of an expected objective which can be interpreted as an infinite realization access setting, see discussion therein) but this formulation is different from ours. Both coincide when the number of trials they consider goes to infinity. Although the problem formulations are different, let us comment further on some additional differences in our results.

2. **Assumptions.** They assume linear realizability of the utility function $F$ with known feature vectors (Assumption 4, p. 17 therein). Our setting differs for two reasons: (1) We do not approximate the utility function itself but rather the occupancy measure and (2) we train a neural network to learn an occupancy measure approximation by maximizing a log-likelihood loss. In our case, our analog (similar but different in formulation and nature) assumption would be our function approximation class regularity assumption (Assumption 1). We do not suppose access to feature vectors which are given. Nevertheless, we do suppose that we can solve the log-likelihood optimization problem to optimality (which is approximated in practice and widely used among practitioners).

3. **Algorithm.** The algorithm they use is model-based, they repeatedly solve a regression problem to approximate the utility function $F$ using samples and use optimism for ensuring sufficient exploration. Our policy gradient algorithm is model-free and we rather rely on MLE for approximating occupancy measures rather than regression.

4. **Analysis.** Under concavity of the utility function, we provide a last iterate global optimality guarantee whereas Mutti et al. (2023) establish an average regret guarantee which is different in nature. Their proof relies on a reduction to an online learning once-per-episode framework. Our proof ideas are different: We combine a gradient optimization analysis exploiting hidden convexity with a statistical complexity analysis for occupancy measure estimation. Overall, our results combine optimization and statistical guarantees whereas their results focus purely on the statistical complexity (as their problem is computationally hard).

**About hardness of occupancy measure estimation.** We comment here on one of the challenges discussed in the main part as for estimating the occupancy measure. An occupancy measure induced

by a policy $\pi$ satisfies the identity $\lambda^\pi(s,a) = \mu_0(s,a) + \gamma \sum_{s' \in \mathcal{S}, a' \in \mathcal{A}} \mathcal{P}(s|s',a')\pi(a|s')\lambda^\pi(s',a')$ where $\mu_0$ is the initial state action distribution. Notice that the sum is not over the next action $s$ in transition kernel $P$ but rather the 'backward' state actions $(s',a')$. In contrast, an action value function in standard RL rather satisfies a 'forward' Bellman equation. In contrast to the standard Bellman equation which can be written using an expectation and leads to a sampled version of the Bellman fixed point equation, the equation satisfied by the occupancy measure cannot be written under an expectation form and does not naturally lead to any stochastic algorithm. This issue is recognized in the literature in Huang et al. (2023) (see also Hallak & Mannor (2017)).

## B  OCCUPANCY MEASURES IN LOW-RANK MDPS

In this section, we show that occupancy measures have a linear structure in the so-called density features in low-rank MDPs. We provide a proof for completeness. Similar results were established in Lemma 16, 17 in Huang et al. (2023) for the finite-horizon setting. Throughout this section, we use the same notations as in the main part of this paper.

**Definition B.1** (Low-rank MDPs). *An MDP is said to be low-rank with dimension $d \geq 1$ if there exists a feature map $\phi : \mathcal{S} \times \mathcal{A} \to \mathbb{R}^d$ and there exist $d$ unknown measures $(\mu_1, \cdots, \mu_d)$ over the state space $\mathcal{S}$ such that for every states $(s,s') \in \mathcal{S}$ and every action $a \in \mathcal{A}$ it holds that*

$$P(s'|s,a) = \langle \phi(s,a), \mu(s') \rangle, \tag{12}$$

*with $\|\phi\|_\infty \leq 1$ without loss of generality.*

Before stating the result, recall that for any policy $\pi \in \Pi$, a state-occupancy measure is defined for every state $s \in \mathcal{S}$ as follows:

$$d^\pi(s) := \sum_{t=0}^\infty \gamma^t \mathbb{P}_{\rho,\pi}(s_t = s). \tag{13}$$

**Lemma 1.** *Consider a low-rank infinite horizon discounted MDP. Then, for any policy $\pi \in \Pi$, there exists a vector $\omega_\pi \in \mathbb{R}^d$ such that the state-action occupancy measure $d^\pi$ induced by the policy $\pi$ satisfies for any state $s \in \mathcal{S}$,*

$$d^\pi(s) = \rho(s) + \langle \omega_\pi, \mu(s) \rangle, \tag{14}$$

*where we use the notation $\mu(s) := (\mu_1(s), \cdots, \mu_d(s))^T$.*

*Proof.* Let $\pi \in \Pi$. It follows from the definition of the state-occupancy measure $d^\pi$ induced by the policy $\pi$ that it satisfies the following (backward) Bellman flow equation for every state $s \in \mathcal{S}$:

$$d^\pi(s) = \rho(s) + \gamma \sum_{s' \in \mathcal{S}, a' \in \mathcal{A}} P(s|s',a')\pi(a'|s')d^\pi(s'). \tag{15}$$

Using the definition of a low-rank MDP and (12) in particular, we obtain:

$$d^\pi(s) = \rho_0(s) + \gamma \sum_{s' \in \mathcal{S}, a' \in \mathcal{A}} \langle \phi(s',a'), \mu(s) \rangle \pi(a'|s')d^\pi(s') \tag{16}$$

$$= \rho_0(s) + \left\langle \gamma \sum_{s' \in \mathcal{S}, a' \in \mathcal{A}} \phi(s',a')\pi(a'|s')d^\pi(s'), \mu(s) \right\rangle \tag{17}$$

$$= \rho_0(s) + \langle \omega_\pi, \phi(s) \rangle, \tag{18}$$

where we define $\omega_\pi := \gamma \sum_{s' \in \mathcal{S}, a' \in \mathcal{A}} \phi(s',a')\pi(a'|s')d^\pi(s')$. $\qquad\square$

## C  EXAMPLES OF NONCONCAVE RLGU PROBLEMS

Nonconvexity is ubiquitous in real-world applications and we provide below a few examples where it naturally arises beyond the standard convex RL examples in the literature. First of all, we would like to mention risk-sensitive RL with non-convex risk measures inspired by Cumulative Prospect

Theory (CPT) (with S-shaped utility curves). Nonconvex criteria are important for modeling human decisions. See e.g. (Lin & Marcus, 2013; Lin et al., 2018) for a discussion about their relevance and importance. See also Remark 1 and figure 2 p. 3 in Prashanth et al. (2016).

Applications include for instance:

- **Robotics control:** in control tasks, it is common to deal with nonconvex objectives such as minimizing energy consumption while achieving a task or maximizing the success rate of a manipulation task.
- **Portfolio Management:** Utility functions in finance may be non-convex due to risk measures or transaction costs for example.
- **Traffic Control:** RL can be used to optimize traffic flow and minimize congestion. The utility function may involve non-convex terms such as travel time, queue lengths, and safety constraints.
- **Supply Chain Management:** RL can be applied to inventory control, pricing, and logistics optimization. The utility function may include non-convex components such as demand forecasting, supply chain disruptions, and dynamic pricing.

We leave the experimental investigation of those applications for future work. We hope our work will foster more research in this direction.

# D    PROOFS FOR SECTION 4

## D.1    STATE SAMPLING FOR MLE

In this section, we briefly discuss how to sample states following the (normalized) state occupancy $d^{\pi_\theta}$ for a given policy $\pi_\theta$. In particular, these states are used for the MLE procedure described in section 3.2. The idea consists in sampling states following the transition kernel $\mathcal{P}$ and the policy $\pi_\theta$ for a random horizon following a geometric distribution of parameter $\gamma$ where $\gamma$ is the discount factor, starting from a state drawn from the initial distribution. The detailed sampling procedure is described in Algorithm 2, borrowed and adapted from Yuan et al. (2023) (Algorithm 3 p. 22) which provides a clear presentation of the idea as well as a simple supporting proof (see Lemma 4 p. 23 therein). This procedure has been commonly used in the literature, see e.g. Algorithm 1 p. 30 and Algorithm 3 p. 34 in Agarwal et al. (2021).

---

**Algorithm 2** Sampler for $s \sim d^{\pi_\theta}_\rho$

---

1: **Input:** Initial state distribution $\rho$, policy $\pi_\theta$, discount factor $\gamma \in [0, 1)$
2: Initialize $s_0 \sim \rho, a_0 \sim \pi_\theta(\cdot|s_0)$, time step $h, t = 0$, variable $X = 1$
3: **while** $X = 1$ **do**
4:    **With probability** $\gamma$**:**
5:    Sample $s_{h+1} \sim \mathcal{P}(\cdot \mid s_h, a_h)$
6:    Sample $a_{h+1} \sim \pi_\theta(\cdot|s_{h+1})$
7:    $h \leftarrow h + 1$
8:    **EndWith**
9:    **Otherwise with probability** $1 - \gamma$**:**
10:    $X = 0$    (Accept $s_h$)
11:    **EndOtherwise**
12: **end while**
13: **Return:** $s_h$

---

## D.2    PROOF OF PROPOSITION 1

Proposition 1 and its proof are largely based on the work of Huang et al. (2023): We follow and reproduce their proof strategy here. Since the latter paper deals with a more complex setting that does not exactly fit our current focus, we provide a proof for clarity and completeness.

We start by defining the concept of $l_1$ optimistic cover. This cover will be immediately useful to quantify the complexity of our (possibly infinite) approximating function class $\Gamma$ defined in (7).

In the following, we denote by $\{\mathcal{X} \to \mathbb{R}\}$ the set of functions defined on $\mathcal{X}$ with values in $\mathbb{R}$.

**Definition D.1** (Definition 3 in Huang et al. (2023)). *For a given function class $\Lambda \subseteq \Delta(\mathcal{X})$, the function class $\bar{\Lambda} \subseteq (\mathcal{X} \to \mathbb{R})$ is said to be an $l_1$ optimistic cover of $\Lambda$ with scale $\kappa > 0$ if:*

$$\forall \lambda \in \Lambda, \quad \exists \bar{\lambda} \in \bar{\Lambda} \quad s.t. \quad \|\lambda - \bar{\lambda}\|_1 \leq \kappa, \quad and \quad \lambda(x) \leq \bar{\lambda}(x), \forall x \in \mathcal{X}. \tag{19}$$

**Remark 4.** *Notice that $\bar{\Lambda}$ does not need to be a set containing only probability distributions if $\Lambda$ is a set of probability distributions, namely the set of (normalized) occupancy measures as we will be considering in the rest of this section.*

We now provide a general statistical guarantee for the maximum likelihood estimator (MLE) defined in (8) supposing we have access to an optimistic cover of the space of distributions used for computing the MLE estimator.

**Proposition 2** (Lemma 12 in Huang et al. (2023)). *Let $\mathcal{D} := \{x_i\}_{i=1}^n$ be a dataset of state-action pairs drawn i.i.d from some fixed probability distribution $\lambda^* \in \Delta(\mathcal{X})$. Let $\Lambda \subseteq \Delta(\mathcal{X})$ be a function class such that:*

*(i) (realizability) $\lambda^* \in \Lambda$,*

*(ii) (probability distribution class) $\forall \lambda \in \Lambda, \lambda \in \Delta(\mathcal{X})$,*

*(iii) (covering) $\Lambda$ has a finite $l_1$-optimistic cover $\bar{\Lambda} \subseteq \{\mathcal{X} \to \mathbb{R}_{\geq 0}\}$ with scale $\kappa$ (see Definition D.1).*

*Then, for any $\delta > 0$, we have with probability at least $1 - \delta$,*

$$\|\hat{\lambda} - \lambda^*\|_1 \leq \kappa + \sqrt{\frac{12 \log\left(\frac{|\bar{\Lambda}|}{\delta}\right)}{n}} + 6\kappa, \tag{20}$$

*where $\hat{\lambda}$ is the MLE estimator defined in (8) computed using the dataset $\mathcal{D}$ and $|\bar{\Lambda}|$ is the cardinality of the finite cover $\bar{\Lambda}$.*

In view of using Proposition 2, the next lemma constructs an $l_1$ optimistic cover for the function approximation class $\Lambda$ used to computed the MLE. For the reader's convenience, we recall that

$$\Lambda := \{p_\omega : \omega \in \Omega \subseteq \mathbb{R}^d, p_\omega \in \Delta(\mathcal{X})\}. \tag{21}$$

**Lemma 2.** *Let Assumption 1 hold. Then there exists a finite $l_1$-optimistic cover $\bar{\Lambda} \subseteq \{\mathcal{X} \to \mathbb{R}_{\geq 0}\}$ of the function class $\Lambda$ with scale $\kappa > 0$ and size at most $2\lceil \frac{B_\omega B_L}{\kappa} \rceil^m$ where $m$ is the dimension of the parameter space $\Omega \subseteq \mathbb{R}^m$.*

*Proof.* The proof follows the same lines as the proof of Lemma 22 p. 41 in Huang et al. (2023). Let $\lambda \in \Lambda$, i.e., $\lambda = p_\omega$ for some $\omega \in \Omega$. Let $\kappa' > 0$. Define the set $\mathcal{B}(\omega, \kappa') := \kappa'\lfloor \frac{\omega}{\kappa'} \rfloor + [0, \kappa']^m$ which is a cubic $\kappa'$-neighborhood of the point $\omega \in \Omega$. Now define the function $f_\omega$ for every $x \in \mathcal{X}$ as follows:

$$f_\omega(x) := \max_{\bar{\omega} \in \mathcal{B}(\omega, \kappa')} p_{\bar{\omega}}(x). \tag{22}$$

By construction, we immediately have $f_\omega(x) \geq p_\omega(x) \geq 0$. Note that $f_\omega$ might not be a probability distribution though. Then using Assumption 1 we also have

$$\|f_\omega - p_\omega\|_1 = \int |f_\omega(x) - p_\omega(x)| dx$$

$$= \int |\max_{\bar{\omega} \in \mathcal{B}(\omega, \kappa')} p_{\bar{\omega}}(x) - p_\omega(x)| dx$$

$$\leq \int \max_{\bar{\omega} \in \mathcal{B}(\omega, \kappa')} |p_{\bar{\omega}}(x) - p_\omega(x)| dx \leq \int \max_{\bar{\omega} \in \mathcal{B}(\omega, \kappa')} |L(x)| \cdot \|\bar{\omega} - \omega\|_\infty dx \leq B_L \kappa'.$$

$$\tag{23}$$

To conclude, we observe that there are at most $2\left\lceil\frac{B_\omega}{\kappa'}\right\rceil^m$ unique functions in the $l_1$-optimistic cover $\bar{\Lambda}$ of $\Lambda$ which is of scale $B_L\kappa'$. Setting $\kappa' = \frac{\kappa}{B_L}$ concludes the proof. $\qquad\square$

**End of Proof of Proposition 1.** We conclude the proof by using Proposition 2 together with Lemma 2 above, choosing a scale $\kappa = \frac{1}{n}$ where $n$ is the number of samples used for computing the MLE and plugging $|\bar{\Lambda}| \leq 2\lceil B_\omega B_L n\rceil^m$. We obtain after simple upper-bounding inequalities,

$$\|\hat{d}^{\pi_\theta} - d^{\pi_\theta}\|_1 \leq 6\sqrt{\frac{12\,m\log\left(\frac{2\lceil B_\omega B_L n\rceil}{\delta}\right)}{n}}\,. \tag{24}$$

### D.3  PROOF OF THEOREM 1

The proof follows similar lines to the proof of Theorem 5.4 in Barakat et al. (2023). However, our occupancy measure estimation procedure is different in the present case. We provide a full proof here for completeness.

We introduce the shorthand notation $\bar{g}_t := \frac{1}{N}\sum_{i=1}^N g(\tau_t^{(i)}, \theta_t, r_t)$ for this proof. Using the smoothness of the objective function $\theta \mapsto F(\lambda(\theta))$ (see Lemma 3 in Appendix D.5) and the update rule of the sequence $(\theta_t)$, we have

$$F(\lambda(\theta_{t+1})) \geq F(\lambda(\theta_t)) + \langle\nabla_\theta F(\lambda(\theta_t)), \theta_{t+1} - \theta_t\rangle - \frac{L_\theta}{2}\|\theta_{t+1} - \theta_t\|^2$$

$$= F(\lambda(\theta_t)) + \alpha\langle\nabla_\theta F(\lambda(\theta_t)), \bar{g}_t\rangle - \frac{L_\theta\alpha^2}{2}\|\bar{g}_t\|^2$$

$$= F(\lambda(\theta_t)) + \alpha\langle\nabla_\theta F(\lambda(\theta_t)) - \bar{g}_t, \bar{g}_t\rangle + \alpha\left(1 - \frac{L_\theta\alpha}{2}\right)\|\bar{g}_t\|^2$$

$$\geq F(\lambda(\theta_t)) - \frac{\alpha}{2}\|\nabla_\theta F(\lambda(\theta_t)) - \bar{g}_t\|^2 - \frac{\alpha}{2}\|\bar{g}_t\|^2 + \alpha\left(1 - \frac{L_\theta\alpha}{2}\right)\|\bar{g}_t\|^2$$

$$= F(\lambda(\theta_t)) - \frac{\alpha}{2}\|\nabla_\theta F(\lambda(\theta_t)) - \bar{g}_t\|^2 + \frac{\alpha}{2}(1 - L_\theta\alpha)\|\bar{g}_t\|^2$$

$$\overset{(i)}{\geq} F(\lambda(\theta_t)) - \frac{\alpha}{2}\|\nabla_\theta F(\lambda(\theta_t)) - \bar{g}_t\|^2 + \frac{\alpha}{4}\|\bar{g}_t\|^2$$

$$= F(\lambda(\theta_t)) - \frac{\alpha}{2}\|\nabla_\theta F(\lambda(\theta_t)) - \bar{g}_t\|^2 + \frac{\alpha}{8}\|\bar{g}_t\|^2 + \frac{\alpha}{8}\|\bar{g}_t\|^2$$

$$\overset{(ii)}{\geq} F(\lambda(\theta_t)) + \frac{\alpha}{16}\|\nabla_\theta F(\lambda(\theta_t))\|^2 - \frac{5}{8}\alpha\|\nabla_\theta F(\lambda(\theta_t)) - \bar{g}_t\|^2 + \frac{\alpha}{8}\|\bar{g}_t\|^2, \tag{25}$$

where (i) follows from the condition $\alpha \leq 1/2L_\theta$ and (ii) from $\frac{1}{2}\|\nabla_\theta F(\lambda(\theta_t))\|^2 \leq \|\bar{g}_t\|^2 + \|\nabla_\theta F(\lambda(\theta_t)) - \bar{g}_t\|^2$.

We now control the last error term in the above inequality in expectation. Recalling that $\nabla_\theta F(\lambda(\theta)) = \nabla_\theta V^{\pi_\theta}(r)|_{r=\nabla_\lambda F(\lambda(\theta))}$ for any $\theta \in \mathbb{R}^d$, we have

$$\mathbb{E}[\|\nabla_\theta F(\lambda(\theta_t)) - \bar{g}_t\|^2] = \mathbb{E}[\|\nabla_\theta V^{\pi_\theta}(r)_{r=\nabla_\lambda F(\nabla(\theta_t))} - \bar{g}_t\|^2]$$

$$\leq 2\mathbb{E}[\|\nabla_\theta V^{\pi_\theta}(r)|_{r=\nabla_\lambda F(\lambda(\theta_t))} - \nabla_\theta V^{\pi_\theta}(r)|_{r=\nabla_\lambda F(\hat{\lambda}_t)}\|^2] + 2\mathbb{E}[\|\nabla_\theta V^{\pi_\theta}(r)|_{r=\nabla_\lambda F(\hat{\lambda}_t)} - \bar{g}_t\|^2]\,. \tag{26}$$

Now, we upper bound each one of the two terms above separately. For convenience, we introduce the notations $r_t := \nabla_\lambda F(\lambda(\theta_t))$ and $\hat{r}_t := \nabla_\lambda F(\hat{\lambda}_t)$.

**Upper bound of the term** $\mathbb{E}[\|\nabla_\theta V^{\pi_\theta}(r_t) - \nabla_\theta V^{\pi_\theta}(\hat{r}_t)\|^2]$ **in** (26)**.** Using the policy gradient theorem (see (4)) yields

$$\nabla_\theta V^{\pi_\theta}(r_t) - \nabla_\theta V^{\pi_\theta}(\hat{r}_t) = \mathbb{E}\left[\sum_{t'=0}^{H-1}\gamma^{t'}[\nabla_\lambda F(\lambda(\theta_t))) - \nabla_\lambda F(\hat{\lambda}_t)]_{s_{t'},a_{t'}} \cdot \left(\sum_{h=0}^{t'}\nabla_\theta\log\pi_\theta(a_h, s_h)\right)\right]\,. \tag{27}$$

Notice that the above expectation is only taken w.r.t. the state action pairs in the random trajectory of length $H$. Taking the norm, we obtain

$$\|\nabla_\theta V^{\pi_\theta}(r_t) - \nabla_\theta V^{\pi_\theta}(\hat{r}_t)\|_2 \overset{(a)}{\leq} \mathbb{E}\left[\sum_{t'=0}^{H-1} \gamma^{t'} \|\nabla_\lambda F(\lambda(\theta_t))) - \nabla_\lambda F(\hat{\lambda}_t)\|_\infty \left\|\sum_{h=0}^{t'} \nabla_\theta \log \pi_\theta(a_h, s_h)\right\|_2\right]$$

$$\overset{(b)}{\leq} \mathbb{E}\left[\sum_{t'=0}^{H-1} 2l_\psi(t'+1)\gamma^{t'} \|\nabla_\lambda F(\lambda(\theta_t))) - \nabla_\lambda F(\hat{\lambda}_t)\|_\infty\right]$$

$$\overset{(c)}{\leq} \mathbb{E}\left[\sum_{t'=0}^{H-1} 2l_\psi L_\lambda(t'+1)\gamma^{t'} \|\lambda(\theta_t) - \hat{\lambda}_t\|_2\right]$$

$$\overset{(d)}{\leq} \frac{2l_\psi L_\lambda}{(1-\gamma)^2} \|\lambda(\theta_t) - \hat{\lambda}_t\|_2, \tag{28}$$

where (a) follows from using the triangle inequality together with the definition of the sup norm, (b) uses Lemma 3 (i) in Appendix D.5, (c) is a consequence of Assumption 3 together with the fact that $\|x\|_\infty \leq \|x\|_2$ for any $x \in \mathbb{R}^d$, and (d) stems from the upper bound $\sum_{t'=0}^{H-1}(t'+1)\gamma^{t'} \leq \sum_{t'=0}^{\infty}(t'+1)\gamma^{t'} = \frac{1}{(1-\gamma)^2}$. Hence we have shown that

$$\mathbb{E}[\|\nabla_\theta V^{\pi_\theta}(r_t) - \nabla_\theta V^{\pi_\theta}(\hat{r}_t)\|_2^2] \leq \frac{4l_\psi^2 L_\lambda^2}{(1-\gamma)^4} \mathbb{E}[\|\lambda(\theta_t) - \hat{\lambda}_t\|_2^2]. \tag{29}$$

**Upper bound of the term** $\mathbb{E}[\|\nabla_\theta V^{\pi_\theta}(\hat{r}_t) - \bar{g}_t\|^2]$ **in** (26)**.** Recalling the definition of $\bar{g}_t$, we have

$$\mathbb{E}[\|\nabla_\theta V^{\pi_\theta}(\hat{r}_t) - \bar{g}_t\|^2] = \mathbb{E}\left[\left\|\frac{1}{N}\sum_{i=1}^{N}(\nabla_\theta V^{\pi_\theta}(\hat{r}_t) - g(\tau_t^{(i)}, \theta_t, \hat{r}_t))\right\|^2\right]$$

$$\overset{(a)}{=} \frac{1}{N}\mathbb{E}[\|g(\tau_t^{(i)}, \theta_t, \hat{r}_t) - \nabla_\theta V^{\pi_\theta}(\hat{r}_t)\|^2]$$

$$\overset{(b)}{\leq} \frac{1}{N}\mathbb{E}[\|g(\tau_t^{(i)}, \theta_t, \hat{r}_t)\|^2]$$

$$\overset{(c)}{\leq} \frac{4l_\lambda^2 l_\psi^2}{(1-\gamma)^4 N}, \tag{30}$$

where (a) follows from the fact that the expectation of $g(\tau_t^{(i)}, \theta_t, \hat{r}_t)$ w.r.t. the random trajectory $\tau_t^{(i)}$ conditioned on $\theta_t$ and $\hat{r}_t$ is precisely given by $\nabla_\theta V^{\pi_\theta}(\hat{r}_t)$ by the policy gradient theorem (see (4)), notice also that all the $N$ trajectories are drawn i.i.d. As for (b), use the fact that the variance of a random variable is upper bounded by its second moment. Finally (c) stems from using the expression of $g(\tau_t^{(i)}, \theta_t, \hat{r}_t)$ in (6) together with Assumptions 2, 3 and Lemma 3 (i) in Appendix D.5. The proof of this last point follows similar lines to (28).

Combining both the previous upper bounds we have now established above, we obtain

$$\mathbb{E}[\|\nabla_\theta F(\lambda(\theta_t)) - \bar{g}_t\|^2] \leq \frac{\tilde{C}_1}{N} + \tilde{C}_2 \cdot \mathbb{E}[\|\lambda(\theta_t) - \hat{\lambda}_t\|_2^2], \tag{31}$$

where $\tilde{C}_1 := \frac{8l_\lambda^2 l_\psi^2}{(1-\gamma)^4}$ and $\tilde{C}_2 := \frac{8l_\psi^2 L_\lambda^2}{(1-\gamma)^4}$.

**End of Proof of Theorem 1.** We are now ready to conclude the proof of our result. Going back to (25), rearranging the terms and taking expectation, we obtain

$$\mathbb{E}[\|\nabla_\theta F(\lambda(\theta_t))\|^2] \leq \frac{16}{\alpha}\mathbb{E}[F(\lambda(\theta_{t+1})) - F(\lambda(\theta_t))] + 10\,\mathbb{E}[\|\nabla_\theta F(\lambda(\theta_t)) - \bar{g}_t\|^2]. \tag{32}$$

Plugging the bound (31) into the previous inequality, we obtain

$$\mathbb{E}[\|\nabla_\theta F(\lambda(\theta_t))\|^2] \leq \frac{16}{\alpha}\mathbb{E}[F(\lambda(\theta_{t+1})) - F(\lambda(\theta_t))] + \frac{10\tilde{C}_1}{N} + 10\tilde{C}_2 \cdot \mathbb{E}[\|\lambda(\theta_t) - \hat{\lambda}_t\|_2^2], \tag{33}$$

Summing the previous inequality for $t = 1$ to $T$, telescoping the right hand side and using the upper bound $F^\star$ on the objective function leads to

$$\frac{1}{T} \sum_{t=1}^{T} \mathbb{E}[\|\nabla_\theta F(\lambda(\theta_t))\|^2] \leq \frac{16(F^\star - \mathbb{E}[F(\lambda(\theta_1))])}{\alpha T} + \frac{10\tilde{C}_1}{N} + \frac{10\tilde{C}_2}{T} \sum_{t=1}^{T} \mathbb{E}[\|\lambda(\theta_t) - \hat{\lambda}_t\|_2^2]. \quad (34)$$

Setting $C_1 := 10\tilde{C}_1$ and $C_2 := \tilde{C}_2$ gives the desired result.

### D.4 PROOF OF THEOREM 2

The proof of this result borrows some ideas from Zhang et al. (2021) and Barakat et al. (2023). However the algorithm we are analyzing is different and the proof deviates from the aforementioned results accordingly.

**Remark 5.** *A different technical analysis can be found in Fatkhullin et al. (2023) by considering a particular case of their theorem 5 dealing with stochastic optimization under hidden convexity. However, their general setting is not focused on our specific RLGU setting using policy parametrization and specifying the assumptions needed as a consequence. More importantly, we are considering a context in which unknown occupancy measures are approximated via function approximation using relevant collected state samples and our theorem accounts for the induced error. In contrast, Fatkhullin et al. (2023) assume access to an unbiased estimate of the gradient of the utility function which is not readily available in our RLGU setting since occupancy measures are unknown and estimated via function approximation with a supporting sample complexity guarantee. Besides these differences, we conduct a different analysis which is rather inspired by the proofs in Zhang et al. (2021) and Barakat et al. (2023) as previously mentioned.*

It follows from smoothness of the objective function $\theta \mapsto F(\lambda(\theta))$ (see (25)) that for every iteration $t$,

$$F(\lambda(\theta_{t+1})) \geq F(\lambda(\theta_t)) + \frac{\alpha}{16} \|\nabla_\theta F(\lambda(\theta_t))\|^2 - \frac{5}{8}\alpha\|\nabla_\theta F(\lambda(\theta_t)) - \bar{g}_t\|^2 + \frac{\alpha}{8}\|\bar{g}_t\|^2. \quad (35)$$

For any $\eta < \bar{\eta}$, the concavity reparametrization assumption implies that $(1 - \eta)\lambda(\theta_t) + \eta\lambda(\theta^*) \in \mathcal{V}_{\lambda(\theta_t)}$ and therefore we have

$$\theta_\eta := (\lambda|_{\mathcal{U}_{\theta_t}})^{-1}((1 - \eta)\lambda(\theta_t) + \eta\lambda(\theta^*)) \in \mathcal{U}_{\theta_t}. \quad (36)$$

It also follows from the smoothness of the objective function $\theta \mapsto F(\lambda(\theta))$ that

$$F(\lambda(\theta_t)) \geq F(\lambda(\theta_\eta)) - \langle \nabla_\theta F(\lambda(\theta_t)), \theta_\eta - \theta_t \rangle - \frac{L_\theta}{2}\|\theta_\eta - \theta_t\|^2. \quad (37)$$

Combining (35) and (37), we obtain

$$F(\lambda(\theta_{t+1})) \geq F(\lambda(\theta_\eta)) - \langle \nabla_\theta F(\lambda(\theta_t)), \theta_\eta - \theta_t \rangle - \frac{L_\theta}{2}\|\theta_\eta - \theta_t\|^2$$
$$+ \frac{\alpha}{16}\|\nabla_\theta F(\lambda(\theta_t))\|^2 - \frac{5}{8}\alpha\|\nabla_\theta F(\lambda(\theta_t)) - \bar{g}_t\|^2 + \frac{\alpha}{8}\|\bar{g}_t\|^2. \quad (38)$$

Now, pick $a \leq \frac{1}{16}$, using Young's inequality gives

$$\langle \nabla_\theta F(\lambda(\theta_t)), \theta_\eta - \theta_t \rangle \leq a\alpha\|\nabla_\theta F(\lambda(\theta_t))\|^2 + \frac{1}{a\alpha}\|\theta_\eta - \theta_t\|^2. \quad (39)$$

Plugging this inequality into (38) yields

$$F(\lambda(\theta_{t+1})) \geq F(\lambda(\theta_\eta)) + (\frac{\alpha}{16} - a\alpha)\|\nabla_\theta F(\lambda(\theta_t))\|^2 + \frac{\alpha}{8}\|\bar{g}_t\|^2$$
$$- \left(\frac{L_\theta}{2} + \frac{1}{a\alpha}\right)\|\theta_\eta - \theta_t\|^2 - \frac{5}{8}\alpha\|\nabla_\theta F(\lambda(\theta_t)) - \bar{g}_t\|^2. \quad (40)$$

Therefore, since $a \leq \frac{1}{16}$, we obtain

$$F(\lambda(\theta_{t+1})) \geq F(\lambda(\theta_\eta)) - \left(\frac{L_\theta}{2} + \frac{1}{a\alpha}\right)\|\theta_\eta - \theta_t\|^2 - \frac{5}{8}\alpha\|\nabla_\theta F(\lambda(\theta_t)) - \bar{g}_t\|^2. \quad (41)$$

Using the definition of $\theta_\eta$ and the concavity of $F$ (Assumption 4), we now control each one of the terms $F(\lambda(\theta_\eta))$ and $\|\theta_\eta - \theta_t\|^2$.

(i) By concavity of $F$ (Assumption 4) and using the definition of $\theta_\eta$, we have

$$F(\lambda(\theta_\eta)) = F((1-\eta)\lambda(\theta_t) + \eta\lambda(\theta^*)) \geq (1-\eta)F(\lambda(\theta_t)) + \eta F(\lambda(\theta^*)). \qquad (42)$$

(ii) Using the uniform Lipschitzness of the inverse mapping $(\lambda|_{\mathcal{U}_{\theta_t}})^{-1}$ (see Assumption 5), we have

$$\begin{aligned}
\|\theta_\eta - \theta_t\|^2 &= \|(\lambda|_{\mathcal{U}_{\theta_t}})^{-1}((1-\eta)\lambda(\theta_t) + \eta\lambda(\theta^*)) - (\lambda|_{\mathcal{U}_{\theta_t}})^{-1}(\lambda(\theta_t))\|^2 \\
&\leq l_\theta^2 \eta^2 \|\lambda(\theta_t) - \lambda(\theta^*)\|^2 \\
&\leq \frac{4l_\theta^2 \eta^2}{(1-\gamma)^2}.
\end{aligned} \qquad (43)$$

Injecting (42) and (43) into (41) yields

$$F(\lambda(\theta_{t+1})) \geq (1-\eta)F(\lambda(\theta_t)) + \eta F(\lambda(\theta^*)) - \left(\frac{L_\theta}{2} + \frac{1}{a\alpha}\right)\frac{4l_\theta^2}{(1-\gamma)^2}\eta^2 - \frac{5}{8}\alpha\|\nabla_\theta F(\lambda(\theta_t)) - \bar{g}_t\|^2. \qquad (44)$$

Rearranging the above inequality, adding $F^*$ to both sides, taking expectation and using the notation $\delta_t := \mathbb{E}[F^* - F(\lambda(\theta_t))]$, we obtain

$$\delta_{t+1} \leq (1-\eta)\delta_t + \left(\frac{L_\theta}{2} + \frac{1}{a\alpha}\right)\frac{4l_\theta^2}{(1-\gamma)^2}\eta^2 + \frac{5}{8}\alpha\mathbb{E}[\|\nabla_\theta F(\lambda(\theta_t)) - \bar{g}_t\|^2]. \qquad (45)$$

Recall then from (31) that

$$\mathbb{E}[\|\nabla_\theta F(\lambda(\theta_t)) - \bar{g}_t\|^2] \leq \frac{\tilde{C}_1}{N} + \tilde{C}_2 \cdot \mathbb{E}[\|\lambda(\theta_t) - \hat{\lambda}_t\|_2^2]. \qquad (46)$$

Since $\mathbb{E}[\|\lambda(\theta_t) - \hat{\lambda}_t\|_2^2] \leq \epsilon_{\text{MLE}}$ uniformly over the iterations, we get by combining (45) and (46) that

$$\delta_{t+1} \leq (1-\eta)\delta_t + \left(\frac{L_\theta}{2} + \frac{1}{a\alpha}\right)\frac{4l_\theta^2}{(1-\gamma)^2}\eta^2 + \frac{5}{8}\alpha\left(\frac{\tilde{C}_1}{N} + \tilde{C}_2\epsilon_{\text{MLE}}\right). \qquad (47)$$

Finally, unrolling this recursion gives

$$\delta_T \leq (1-\eta)^T\delta_0 + \left(\frac{L_\theta}{2} + \frac{1}{a\alpha}\right)\frac{4l_\theta^2}{(1-\gamma)^2}\eta + \frac{5}{8}\frac{\alpha}{\eta}\left(\frac{\tilde{C}_1}{N} + \tilde{C}_2\epsilon_{\text{MLE}}\right). \qquad (48)$$

### D.5 USEFUL TECHNICAL RESULT

**Lemma 3** (Lemma 5.3, Zhang et al. (2021))**.** *Let Assumptions 2 and 3 hold. Then, the following statements hold:*

(i) $\forall \theta \in \mathbb{R}^d, \forall (s,a) \in \mathcal{S} \times \mathcal{A}, \|\nabla \log \pi_\theta(a|s)\| \leq 2l_\psi, \|\nabla_\theta^2 \log \pi_\theta(a|s)\| \leq 2(L_\psi + l_\psi^2)$, *and* $\|\nabla_\theta F(\lambda(\theta))\| \leq \frac{2l_\psi l_\lambda}{(1-\gamma)^2}$.

(ii) *The objective function* $\theta \mapsto F(\lambda^{\pi_\theta})$ *is* $L_\theta$-*smooth with* $L_\theta = \frac{4L_{\lambda,\infty}l_\psi^2}{(1-\gamma)^4} + \frac{8l_\psi^2 l_\lambda}{(1-\gamma)^3} + \frac{2l_\lambda(L_\psi + l_\psi^2)}{(1-\gamma)^2}$.

## E ADDITIONAL DETAILS FOR EXPERIMENTS

In this section, we provide additional details related to the experiments in this work.

**Hardware configuration.** We conducted experiments on a cluster of Nvidia GPUs with Intel Xeon processors, running on Linux.

**1. Discrete Gridworld Environment.** Figure 4 visualizes our experimenting gridworld environments (Yu et al., 2024). We train each individual agent separately using dense reward and collect the demonstration trajectories with the learned optimal policies.

**Networks architectures.** Details of the Actor and Critic network architectures are provided below:

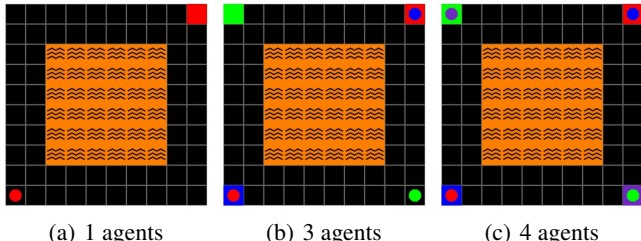

(a) 1 agents      (b) 3 agents      (c) 4 agents

Figure 4: We utilize a 10x10 gridworld environment featuring a central 6x6 area filled with lava and generate scenarios with 1, 3, and 4 agents, where each agent is positioned initially at distinct corners of the grid. The agents' objective is to navigate to the diagonally opposite corner of the grid. Circles indicate the start locations of the agents, and squares with the same color indicate the corresponding goal locations for each agent.

- Actor Network: [Linear(obs_dim, 64), Tanh, Linear(64, action_dim), Softmax]
- Critic Network: [Linear(obs_dim, 64), Tanh, Linear(64, 1)] .

Inside our proposed algorithm, we train a discriminator with the following architectures:

- Discriminator Network: [Linear(obs_dim + action_dim, 64), Tanh, Linear(64, 64), Tanh, Linear(64, 1), Sigmoid]

**Count-based baseline.** Regarding the count-based algorithm which is a vanilla PG algorithm (see Algorithm 3 in Barakat et al. (2023) without occupancy approximation or Zhang et al. (2021) (without variance reduction)), we perform $B$ environmental rollouts and calculate the occupancy measures by counting different state-action pairs and averaging them. This is the simple Monte Carlo estimator for the state occupancy measure computing state frequencies as previously used in Zhang et al. (2021) (see eq. (6) therein) and Barakat et al. (2023) (see eq. (8) therein). The estimator for the state-action occupancy measure $\lambda^{\pi_\theta} = \lambda(\theta)$ (see (1)) truncated at the horizon $H$ is defined as follows:

$$\lambda(\tau) = \sum_{h=0}^{H-1} \gamma^h \delta_{s_h, a_h} \,, \tag{49}$$

where $\tau$ is a trajectory of length $H$ generated by the MDP controlled by the policy $\pi_\theta$ and for every $(s, a) \in \mathcal{S} \times \mathcal{A}$, $\delta_{s,a} \in \mathbb{R}^{|\mathcal{S}| \times |\mathcal{A}|}$ is a vector of the canonical basis of $\mathbb{R}^{|\mathcal{S}| \times |\mathcal{A}|}$, i.e., the vector whose only non-zero entry is the $(s, a)$-th entry which is equal to 1. Figure 5 shows the policy convergence rate over different choices of batch sizes $B$.

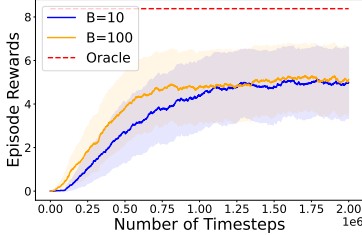

Figure 5: We evaluate varying numbers of samples for computing the occupancy measure in the Gridworld (1-agent case) environment. We observe that the learned policy converges faster with a larger batch size since the occupancy measure estimation is more accurate.

**Occupancy approximation.** For our discrete state environments, we use a softmax parametrization akin to the one introduced in section (3.2). If a finite batch of trajectories does not cover the entire state space, a softmax can be computed either over the support of the state space covered or by assigning low/dummy values to irrelevant unseen states.

**Our multi-agent setting.** We believe that it is natural to consider the multi-agent setting since the dimensionality of the occupancy measure grows exponentially as the number of agents increases. It is easy to demonstrate why the count-based method does not perform well compared to our method. Each agent is controlled by an independent policy, and agents policies are trained together.

**Suboptimal vs optimal demonstrations.** The suboptimal demonstrations have lower episode returns than the optimal ones. The average return of the suboptimal demonstrations is about half of the optimal one.

**Confidence intervals.** Each of our experiments is performed over 5 different random initializations and we plot the mean and variance across all the runs.

**2. Continuous environments**. For the continuous state space environments, we consider the cooperative navigation task of multi-agent particle environment (MPE) (Lowe et al., 2017) and StarCraft Multi-Agent Challenge (SMAC) environment (Samvelyan et al., 2019). MPE is a benchmark for multi-agent RL involving simple physics-based interactions. SMAC is a challenging environment based on the StarCraft II game, used to test multi-agent coordination and strategy. From SMAC, we consider 3sv4z, which features 3 Stalkers (allies) versus 4 Zealots (enemies).

**Discretization of the continuous space for baseline.** We only perform discretization for the MPE environment. The observation of the MPE environment includes the agent's velocity, position, all landmarks' and other agents' relative position wrt it. We basically discretize over the first 4 dimensions of the observation (velocity and position), where each dimension is discretized into 20 bins, and then calculate the occupancy measure.

**Win rate for SMAC.** In our StarCraft task, 3 ally agents need to defeat 4 enemy agents. The win rate measures the probability that the ally agents win. So it is about winning the game: The higher the better.

**Hyperparameter Values.** For both the experimental settings in the paper in Figure 2 and Figure 3, we utilized the following values.

| Parameter | Value |
|:---------:|:-----:|
| epochs | 4 |
| buffer size | 4096 |
| clip | 0.2 |
| learning rate | 1e-4 |

Table 2: Hyperparameters for Figure 2 (navigation task).

| Parameter | MPE | SMAC (3sv4z) |
|:---------:|:---:|:------------:|
| epochs | 10 | 15 |
| buffer size | 1024 | 1024 |
| gain | 0.01 | 0.01 |
| clip | 0.05 | 0.2 |
| learning rate | 1e-3 | 3e-4 |

Table 3: Hyperparameters for Figure 3 (continuous environments).

**Fine-tuning.** For our experiments in this work, we adopt the parameters mentioned in the baseline PPO implementation (Yu et al., 2022, Table 13) and further fine-tune the learning rate parameter to obtain stable and convergence behaviour of the proposed algorithm.

# F   ABOUT FUTURE WORK

We comment here on a few future directions of improvement:

- In our PG algorithm, the estimations of the state occupancy measure need to be relearned for each policy parameter $\theta_t$. We believe a regularized policy optimization approach could lead to a more efficient procedure. Indeed, by enforcing policy parameters to be not too

far from each other, it would allow to reuse estimations of the occupancy measure from previous iterations to obtain better and more reliable estimations.

- The state-occupancy measure can be very complicated and hence difficult to estimate, especially in complex high-dimensional state settings. The use of massively overparametrized neural networks for occupancy measure approximation might therefore be of much help in such complex settings as practice shows that overparametrized neural networks do perform well in general. Establishing theoretical guarantees in this regime is certainly an interesting question to extend our work.

- It would definitely be interesting to conduct experiments in very large scale environments such as DMLab or Atari. Our work makes progress towards solving larger scale real-world RLGU problems and offers a promising approach supported by theoretical guarantees.

