# OpenReview forum: "On the Sample Complexity of a Policy Gradient Algorithm with Occupancy Approximation for General Utility Reinforcement Learning"
_ICLR.cc/2025/Conference — Submitted to ICLR 2025_

### Official Review · Reviewer_MseJ · 2024-11-01

**Soundness:** 3
**Presentation:** 3
**Contribution:** 1
**Rating:** 3
**Confidence:** 4

**Summary:**

This paper propose to approximate the occupancy measure within a function approximation class using maximum likelihood estimation (MLE). Based on this, it develops a policy gradient method for General Utility RL, demonstrates the sample complexity for non-concave and concave utility functions, and shows the experiment results in discrete and continuous state-space settings.

**Strengths:**

This paper offers a  method to estimate occupancy measure potentially for large-scale state-action spaces. It shows the convergence rate for both non-concave and concave utility functions. The paper is well-written and easy to read.

**Weaknesses:**

The only novelty of this paper is the use of MLE in estimating the occupancy measures. Other parts in the method, such as the form of policy gradient for convex RL, are known results. However, prior work such as Barakat et al (2023) have already considered a very similar approach using mean square error (MSE) estimation. The replacement of MSE by MLE seems trivial, and the consequent analysis seems straightforward (MSE and MLE are two common estimation methods, and the only difference seems to be different loss objectives used
in the neural network). Although the authors carried out a detailed analytical comparison between MSE and MLE in Appendix A, the comparison is not very convincing (for example, when comparing the TV bound on the estimation error, the TV bound of MSE is obtained by relaxation and the dependce on |X| is based on the assumption that p* is uniform). There is no numerical comparison between the methods using MLE versus MSE. Overall, I think the novelty and contribution of this paper is quite limited.

**Questions:**

1. Please state the novelty of this paper clearly in the introduction, especially the connection and difference with the prior work that uses MSE for estimating the occupancy measure. Is there any technical challenge in replacing MSE with MLE? How to choose the
function class so that the true occupancy measure can be estimated well, i.e. Assumption 1(ii) is satisfied?

2. The dependency of MSE on |X| is currently shown based on the assumption that p^* is uniform. Can you provide a more rigorous analysis for the general case to support the claim that MSE depends on |X| and suffers from scalability issue?

3. As stated in the paper, MSE method may introduce a scalability issue while MLE does not. In addition to the analytical comparison (as in question 2 above), can you compare MLE and MSE numerically to support this claim?

3. Is there any challenge in the proof of Theorem 1 and Theorem 2? Please highlight the challenges in adapting the analysis in Zhang et al. (2021) to this paper.

---

> ### Author Response · Authors · 2024-11-25
> **Response to reviewer MseJ**
>
> We thank the reviewer for their feedback and for their questions. We are glad that  they found the paper ‘well-written and easy to read.’ We respond to each one of their comments and questions in the following to further clarify our contributions.
>
> **Preliminary points about novelty compared to Barakat et al. 2023.** Let us first recall a few points regarding our contributions compared to that work along the lines of the detailed comparison we provided in the appendix (p. 15-16) and l. 265-267 in the main part.
>
> 1. **Experiments.** They do not provide any simulations testing their algorithm in section 5 for large state action spaces, Fig. 1 therein is only for the tabular setting for a different algorithm. We do provide simulations and compare to the tabular setting (with count-based approaches) to demonstrate the scalability issue.
>
> 2. **Global convergence.** In contrast to our work (see our theorem 2 and corollary 2), they only provide a first-order stationarity guarantee and they do not provide global convergence guarantees;
>
> 3. **Technical analysis.** From the technical viewpoint, our occupancy measure MLE estimation procedure combined with our PG algorithm requires a different theoretical analysis even for our first order stationarity guarantee. Please see appendix D for details.
>
> Due to the length of the discussion, we have delayed it to the appendix while only keeping the salient points of the comparison (namely the advantage of MLE compared to their MSE approach) to avoid interrupting the flow of the paper. We will expand on this discussion in the main part to further clarify our contributions as suggested by the reviewer.
>
> **MSE in Barakat et al. 23 vs MLE (ours) for distribution estimation.** While this might indeed be a simple algorithmic change (and simplicity of the algorithm should perhaps be seen positively), we argue that it is crucial given the main goal and motivation of scaling to larger state action spaces. Our discussion might have given the impression that the assumption that p^* is uniform is needed to showcase our point, it is not. Our discussion regarding this point might have confused the reviewer. We provide clarifications here regarding why the MSE formulation in Barakat et al. 2023 is problematic for occupancy measure approximation regarding scalability and other fundamental aspects. Recall from that paper (see Eq. 11 therein) that the loss for approximating an occupancy measure (induced here by policy $\pi_{\theta}$) via linear regression is $E_{s \sim \rho, a \sim \mathcal{U}(\mathcal{A})}[(\lambda^{\pi_{\theta}}(s,a) - \langle \phi(s,a), \omega \rangle)^2]$  where $\rho$ is the initial state distribution and $\mathcal{U}(\mathcal{A})$ the uniform action distribution. We now make a few points regarding why there are issues with this objective:
>
> 1. **The expected loss is over the initial state distribution (defining the MDP)**. Take the extreme case where we initialize at a single state (note that this is also realistic, e.g. a robot starting at a given deterministic state). Then this means that the expected loss boils down to estimating the occupancy measure only at that state. However we need to estimate it as accurately as possible for all states and there is no reason why the occupancy measure should be supported by the same set of states as the initial distribution (which we should have freedom about). Note also that the occupancy measure itself depends on the initial distribution. In principle, the distribution used for defining the expected loss (over state action pairs) should be different from the initial state distribution defining MDP.

---

> ### Author Response · Authors · 2024-11-25
> **Further discussion about MSE vs MLE**
>
> 2. **Coverage and scalability problem.** Now you may argue that it is enough to take an initial distribution (or just $\rho$ distribution for the expected loss if one assumes it is unrelated to the initial distribution) that just needs to cover the support of the occupancy measure we want to estimate. Note that the occupancy measure is unknown and we want to estimate it so we have a priori no clue about its support. One might then think about just taking the uniform distribution as an initial distribution to be sure to cover the whole state space equally. This choice is problematic for several reasons: (a) First, this introduces a bias: Why would we need to estimate the occupancy measure equally well in all the states if the occupancy measure is concentrated on a specific set of states which is not necessarily the entire state space? (b) Second and most importantly,  if we make such a choice, we have now $\rho_min = 1/|S|$ (say in the discrete state space setting) and the first order stationarity bound in Barakat et al. 23 scales with $1/\rho_{\min}$, this makes the result not scalable to large state spaces. You might argue that we do not need the uniform distribution but just to take a distribution $\rho$ covering the entire state space (not necessarily equally well), i.e. which has a support equal to the entire state space. Then again, this introduces a bias as the loss minimization might focus on states which are irrelevant to the occupancy measure we want to estimate.
>
> 3. An additional point that we want to highlight here and that relates to the shortcomings above is that MSE might not be the best metric for comparing distributions because it focuses on pointwise differences (in our case states or state-action pairs). In our setting, how the weights of the MSE loss are chosen for fitting our probability distribution of interest is important.
>
> **There are a number of shortcomings of using MSE compared to MLE for fitting a probability distribution in general, we summarize them here:**
>
> 1. **Consistency and efficiency.** MLE maximizes the likelihood function, ensuring the estimator is consistent (i.e. converges to the true parameter value as the sample size increases) and asymptotically efficient (i.e. achieves the lowest possible variance among unbiased estimators for large samples). In contrast, as MSE minimizes mean squared errors, it does not guarantee properties like consistency or efficiency unless under specific assumptions on the data such as normality of the errors in which case both coincide. Note that the approach in Barakat et al. 2023 does not fit a Gaussian distribution to the normalized occupancy measure or anything like that. We use favorable statistical properties of MLE (see our proposition 1).
>
> 2. **Sensitivity to scaling.** This is an important point regarding our discussion that also relates in a way to scaling to large state spaces. MLE operates on probabilities and likelihoods, these are normalized and scale-invariant. This makes MLE very suitable for probability distribution fitting in general. MSE is rather better used for point wise estimation in statistics (which is also indirectly used for distribution estimation via estimating parameters such as Gaussian means), MSE depends on the scale of the observations which might make it less robust in some settings. Please see also next point as a follow-up.
>
> 3. **Robustness to outliers.** As MLE models probabilities directly, it might be more robust to outliers depending on the distribution. MSE can be more sensitive to outliers and extreme values as it relies on squared errors which amplify such outliers. In our setting, this is also relevant as we are interested in estimating occupancy measures on large state spaces, this induces small probability values (even extremely small for some of them) and squaring differences makes it worse, this is what we have explained in the appendix with simple calculations showing how the state space size shows up.
>
> 4. **Satisfying distribution constraints.** MLE naturally adapts to the distribution's shape and constraints to satisfy them. In contrast, MSE can lead to estimates that violate distribution constraints such as probability normalization or predicting a negative variance for a normal distribution. Therefore, post-processing might be required to ensure these are satisfied.
>
> We focused on the state space here, similar comments can be made for the choice of the uniform action distribution in the expected loss.
>
> We hope that these clarifications have convinced the reviewer of the fundamental issues with such an MSE approach as well as advantages of ours. We believe that simulations are not even needed to illustrate this (some of these facts which are not necessarily specific to our setting are even general and well-known in statistics). We will add some of these points to our paper to strengthen our motivation for using our approach rather than theirs.

---

> > ### Comment · Reviewer_MseJ · 2024-11-26
> >
> > I would like to thank the reviewer for responding to my comments and questions. Your explaination does clarify some questions I had about the advantage of MLE over MSE and how to position this paper to relevant papers. However, I still think the novelty of the paper is limited, and the paper lacks a rigorous analysis for the general case and numerical comparisons with the MSE approach. Therefore, I decide to keep my rating.

---

> ### Author Response · Authors · 2024-11-25
> **Response to questions**
>
> **Question 1.** We will provide further details as requested by the reviewer. Please see the above comments for an extended discussion.
> As for the choice of the ‘function class so that the true occupancy measure can be estimated well, i.e. Assumption 1(ii) is satisfied’:
>
> - **As we highlight in l. 242-246, state-occupancy measures are linear (or affine in the discounted setting) in density features in low-rank MDPs.** We refer the reader to Appendix B for a proof of this statement. Therefore, in this case, it is natural to approximate occupancy measures via linear function approximation using some density features.
>
> - **As we mention in l. 323, this realizability assumption can be relaxed at the price of incurring a misspecification error.** This assumption allows us to get rid of the approximation error stemming from the fact that the occupancy measure might not belong to our function class. This induced error would depend on the expressivity and richness of the function class. One may relax it by introducing an additional function approximation error and propagate it in our analysis. This would result in a constant approximation error $\epsilon_{\approx}$ (assuming this is the constant upper bound on that error, which is rather standard in the literature, see e.g. Agarwal et al. 2021 ‘On the Theory of Policy Gradient Methods: Optimality, Approximation, and Distribution Shift’) propagating in Proposition 1 and then in our theorems. The propagation can be seen in Theorem 1 where the last term corresponds to the occupancy measure estimation error. We will add a comment about this.
>
> **Questions 2 and 3.** Please see the detailed discussion above.
>
> **Question 4: challenge in the proof of Theorem 1 and Theorem 2 compared to Zhang et al. (2021).**
>
> - Recall first that the analysis of Zhang et al. 2021 is restricted to the tabular setting and their algorithm uses non-scalable count-based estimates of the occupancy measure.
>
> - Our analysis does of course use similar parts of the technical analysis of Zhang et al. 2021 concerning the optimization part (use of hidden convexity with the required policy parametrization assumption and leveraging smoothness of the objective precisely) and we do acknowledge it in the paper and the proofs. Our contributions from the technical viewpoint consist in (a) approximating the occupancy measure, provide statistical guarantees for the MLE approach we follow using prior work, introduce the right assumptions for our problem (see Assumption 1, proposition 1) and (b) propagate the estimation error into the optimization analysis to obtain Theorems 1 and 2 as well as setting up all the parameters (step sizes, number of samples, …) to derive sample complexities.
>
> Thanks again for your review. Please let us know if you have any further comments or questions, we will be happy to address them.

---

> ### Author Response · Authors · 2024-12-01
> **Thank you for your response and follow-up**
>
> Thank you for your response. We want to clarify here that there is 'no rigorous analysis for the general case' to come up with here (as you suggest) as we are not considering a particular case in our discussion above (regarding the MSE approach in prior work) which starts from their general formulation of the problem. We have just clearly shown above that the MSE approach in prior work has fundamental issues even from the initial formulation of the problem of occupancy measure estimation and it has not been tested in practice in prior work.

---

> > ### Author Response · Authors · 2024-12-03
> > **Request to Reviewer**
> >
> > Dear Reviewer,
> >
> > We sincerely appreciate your time and effort to review our work. With the deadline for discussion ending in less than 20 hours, we want to make sure that our responses have addressed all of your concerns. Any additional insight would be very helpful to us. If all of your concerns have been addressed, we would request a reconsideration of your original score.
> >
> > Best,
> > Authors

---

### Official Review · Reviewer_9Est · 2024-11-02

**Soundness:** 3
**Presentation:** 3
**Contribution:** 2
**Rating:** 6
**Confidence:** 3

**Summary:**

The paper investigates the policy gradient algorithm with occupancy approximation for general utility reinforcement learning (GURL). It proposes a policy gradient algorithm that approximates the occupancy measure using maximum likelihood estimation (MLE). The introduction of MLE eliminates the dependency of the estimation error on the state space size, leading to improved performance bounds. The theoretical gain is also corroborated by empirical evidence.

**Strengths:**

**Novelty**: The introduction of MLE for occupancy measure approximation is a novel contribution to the theoretical investigation of GURL. It addresses the limitation of previous count-based approximation and offers advantages over mean square error estimation.

**Clarity**: The paper is well-structured and written, making it easy to read and follow. The main results are clearly highlighted, and the conclusions are succinctly declared. This clarity enhances the reader's comprehension and engagement with the material.

**Significance**: The introduction of MLE for occupancy measure approximation and the theoretical analysis constitute a moderately significant contribution to GURL. It advances the understanding of occupancy measure approximation.

**Quality**: The paper is of high quality, demonstrating rigorous and reliable research. The arguments are well-supported, and the methodologies are robust, contributing to the overall reliability of the results.

**Weaknesses:**

**Predictable Results**: Although the introduction of MLE estimation and corresponding theoretical analysis is a novel and concrete contribution, the state space size independent performance bound is somewhat predictable given the assumptions. This result is not surprising and thus may not add substantial new insights. Additionally, the widespread use of MLE in practice diminishes the practical value of the proposal.

**Unjustified Assumptions**: The paper lacks practical examples that satisfy Assumption 3, which appears quite restrictive. The requirement for utility smoothness is particularly challenging for utilities involving Kullback-Leibler (KL) divergence. Furthermore, Assumption 5 necessitates a bijection relationship between the neighborhood of the softmax policy parameter $\theta$ and the corresponding occupancy measure $\lambda(\theta)$. However, the paper does not provide an example illustrating how this requirement can be fulfilled, leaving the assumption potentially unjustified.

**Questions:**

Can you provide specific types of RLGU problems where these assumptions might hold or discuss how restrictive these assumptions are in practice?

---

> ### Author Response · Authors · 2024-11-25
> **Response to reviewer 9Est**
>
> We thank the reviewer for their positive feedback on our work. We respond to their comments and questions in the following.
>
> > Predictable Results: Although the introduction of MLE estimation and corresponding theoretical analysis is a novel and concrete contribution, the state space size independent performance bound is somewhat predictable given the assumptions. This result is not surprising and thus may not add substantial new insights. Additionally, the widespread use of MLE in practice diminishes the practical value of the proposal.
>
> We thank the reviewer for acknowledging the novelty of our contributions. Leveraging existing statistical estimation error bounds for MLE, we achieve the first 1st order stationarity and global optimality results (with sample complexity guarantees) for solving RLGU in large state action spaces to the best of our knowledge. Estimating occupancy measures using function approximation is not common in the RL literature even if MLE is standard in statistics. We also stress the advantage of using MLE over prior work which proposed a MSE approach which is not well-suited for distribution estimation and which has a number of limitations as it is introduced in Barakat et al. 2023, please see our response to reviewer MseJ for a detailed discussion.
>
> **Assumption 3 (General utility smoothness):**
> - This assumption is quite standard in the RLGU literature: see e.g. Hazan et al. ICML 2019, Zhang et al. NeurIPS 2020, NeurIPS 2021, AAAI 2022, Barakat et al. ICML 2023, Ying et al. AAAI 2023, NeurIPS 2023.
> - The assumption is satisfied for several functions of interest including the standard RL setting, quadratic objectives (recall that the utility variables are confined to the simplex) and more generally for any smooth objective. While the KL divergence is not strictly speaking smooth, the smoothed KL which only adds a small numerical constant inside the log term is actually smooth (as it allows to get rid of the singularity at the boundary of the simplex). The smoothed entropy defined as $H_{\sigma}(x) = - x log(x + \sigma)$ is $\frac{1}{2\sigma}$-smooth w.r.t. the infinity norm and has been used in the RLGU literature (and beyond), we refer the reader to the work of Hazan et al. 2019 (see e.g. Lemma 4.3 therein). Overall this assumption captures most of the problems of interest in RLGU including pure exploration (using the smoothed entropy), learning from demonstrations (using the smoothed KL) as well as standard linear RL and CMDPs. Other entropic measures or $l^2$ losses are also possible.
>
> **Assumption 5:**
> - This assumption is also commonly used in the RLGU literature, in all the aforementioned works that we mentioned for Assumption 3 (see also Ying et al. 2024 ‘Policy-based Primal-Dual Methods for Concave CMDP with Variance Reduction’).
> - It is satisfied for instance for tabular policy parameterizations. However, as we mention in the paper, this assumption is difficult to verify in general and this is also acknowledged by prior work. We note that it is only a local assumption though. Trying to relax such an assumption is an interesting question, we leave this for future work.
> - Note that this assumption is not needed for our first order stationarity results (Theorem 1 and Corollary 1). It is only required for Theorem 2 and Corollary 2.
>
>
> **Can you provide specific types of RLGU problems where these assumptions might hold or discuss how restrictive these assumptions are in practice?**
>
> Please see our responses above.  All our assumptions are discussed in the paper and examples as well as references are provided, except for the smoothness one (Assumption 3). Concerning this standard assumption, we will add a brief discussion along the lines of our response above for completeness.
>
> Thank you for your review and your positive assessment of our work. Please let us know if you have any further questions.

---

### Official Review · Reviewer_Kpay · 2024-11-03

**Soundness:** 3
**Presentation:** 3
**Contribution:** 2
**Rating:** 6
**Confidence:** 3

**Summary:**

This paper proposes a policy gradient algorithm for general utility reinforcement learning where an estimation of occupancy measure is involved to address the challenge caused by large state-action spaces. Provided theoretical analysis covers both statistical and optimization aspects. Additionally, numerical results are presented to demonstrate the algorithm's effectiveness in tasks involving large state-action spaces.

**Strengths:**

This paper identifies a challenge in General Utility Reinforcement Learning/Convex Reinforcement Learning caused by large state-action spaces and proposes a method to address it. The motivation is well-founded, and the proposed method is sound and natural, i.e. utilizing a maximum likelihood estimator to estimate the occupancy measure. The theoretical analysis is comprehensive, and the paper includes a reasonably thorough comparison with closely related existing works in the Appendix. The structure is logical, with ideas presented in a clear and coherent manner.

**Weaknesses:**

1. The primary weakness seems to be the limited novelty of the idea. Although the authors provide a comparison with a related work [1] in the Appendix, the advantages of the proposed method over [1] are not immediately clear. For example, it would be beneficial to include a more detailed discussion of the assumptions required in this work versus those in [1], clarifying which are weaker, stronger, or more practical for specific problem settings. Additionally, highlighting the advantages of the proposed model-free algorithm over the model-based approach in [1] would also be helpful. I think these would significantly improve the paper by more clearly demonstrating its unique contributions.

2. Based on the content in the Appendix and my observations, the technical theoretical analysis appears to draw upon existing results. The statistical analysis is derived from [2], while the optimization analysis is based on [3], with minor adaptations to handle estimation error. However, this is not a critical issue if the authors address the concerns outlined in the Weakness 1, as this would help establish the paper’s distinct contributions more effectively.

[1] Mutti, M., De Santi, R., De Bartolomeis, P., & Restelli, M. (2023). Convex reinforcement learning in finite trials. Journal of Machine Learning Research, 24(250), 1-42.

[2] Huang, A., Chen, J., & Jiang, N. (2023, July). Reinforcement learning in low-rank mdps with density features. In International Conference on Machine Learning (pp. 13710-13752). PMLR.

[3] Barakat, A., Fatkhullin, I., & He, N. (2023, July). Reinforcement learning with general utilities: Simpler variance reduction and large state-action space. In International Conference on Machine Learning (pp. 1753-1800). PMLR.

**Questions:**

1. It would be helpful if the authors could clarify the additional assumptions or new technical tools required for proving global convergence compared to [1].

2. Compared to [2], I am wondering if there exists any sacrifice in the results when you prove a last-iterate global convergence, e.g. looser upper bound, more restrictive assumptions, etc.

3. In Theorem 2, does $\epsilon_{MLE}$ implicitly depend on the number of iterations, $T$? Since the estimation errors for occupancy must be uniformly controlled across all iterations, it seems intuitive that $\epsilon_{MLE}$ might depend on $T$. Explicitly stating this dependence would clarify its impact on Corollary 2.

4.  Also, I think $\epsilon_{MLE}$ should depend on the size of the policy space because $\theta_t$ is not fixed in each iteration. If I understand it correctly,  you essentially need $sup_{t=1,...,T} \sup_{\theta_t\in \Theta}$(Estimation error at $t$) $<\epsilon_{MLE}$. I suggest to discuss more on this as the dependence on the size of policy space typically appears in the model-free methods.

[1] Barakat, A., Fatkhullin, I., & He, N. (2023, July). Reinforcement learning with general utilities: Simpler variance reduction and large state-action space. In International Conference on Machine Learning (pp. 1753-1800). PMLR.

[2] Mutti, M., De Santi, R., De Bartolomeis, P., & Restelli, M. (2023). Convex reinforcement learning in finite trials. Journal of Machine Learning Research, 24(250), 1-42.

---

> ### Author Response · Authors · 2024-11-25
> **Response to reviewer Kpay**
>
> We thank the reviewer for their time and feedback on our work. We appreciate that the reviewer finds that our 'theoretical analysis is comprehensive' and that 'structure is logical, with ideas presented in a clear and coherent manner'. We reply to their comments and questions in what follows.
>
> **Comparison to Mutti et al. 2023.**
> To complement our detailed comparison in appendix A p. 16 (along several dimensions including the problem formulation, the assumptions, the algorithm and the analysis) we provide a few additional points to answer the reviewer’s follow-up questions.
>
> **Advantages of our analysis/approach.** Let us list first a few advantages/differences w.r.t. the aforementioned work:
>
> 1. **Policy parameterization.** We consider policy parameterization instead of tabular policies which results in a practical algorithm. We do require a strong assumption though (Assumption 5) to obtain our global optimality result.
>
> 2. **No planning oracle access.** We do not suppose access to a planning oracle able to solve any convex MDP efficiently. This is precisely the point of our optimization guarantees for our PG algorithm which updates policies incrementally. We discuss in particular how to set up the step size and other parameters to obtain our results. Nevertheless, we point out here that Mutti et al. 2023 address a slightly different (finite-trial) convex RL problem which is computationally intractable. Our problem coincides with their infinite trial variant of the problem.
>
> 3. **No access required to feature vectors for function approximation.** We learn occupancy measure approximations rather than supposing access to a set of features to approximate utilities (i.e. $F(\lambda)$ in our notations), i.e. we do not suppose access to a set of basis feature vectors for our approximation.
>
> 4. **Model-free and no dependence on state action space size.** We do not estimate the transition kernel, our algorithm is model-free as we only require access to sampled trajectories. In particular, we do not require to go through the entire state action space to estimate each entry of the transition kernel. More importantly, our performance bounds do not have dependence on the state action space sizes as their regret bound.
>
> **Advantages of our model-free PG algorithm vs their model-based algorithm.** We inherit the usual advantages of model-free vs model-based algorithms. We list some of these:
>
> 1. **Implementation.** Model-free are often simpler to implement because they do not require learning or using a model of the environment. Our algorithm directly focuses on learning a policy (even if we do also approximate the occupancy measures but not the transitions themselves like in model-based approaches). We do not require to model dynamics explicitly and we require less assumptions on the environment’s dynamics.
>
> 2. **Robustness.** Inaccuracies in environment modeling propagate to policy optimization and can significantly degrade performance.  Model-free methods directly learn policies from interaction with the environment.
>
> 3. **Complex and high-dimensional environments.** PG methods are particularly suitable for such settings. Estimating a model accurately in such settings can be difficult.
>
> Of course, model-based methods also have advantages over model-free ones. For instance, model-free methods generally require more samples (note this is the case when comparing the result of Mutti et al. 2023 and ours) and model-based methods can also guide exploration.
>
> **Assumptions.** Besides the points above, we provide a few comments regarding assumptions:
> 1. Some of our assumptions are quite similar. For instance, Mutti et al. 2023 assume linear realizability of the utility function $F$ with known feature vectors (Assumption 4, p. 17 therein) and then assume access to a regression problem solver with cross-entropy loss to approximate the utility function. We rather have a similar but different function approximation class regularity assumption (Assumption 1) and we suppose access to an optimizer which solves our log-likelihood loss maximization problem to approximate occupancy measures (see Eq. (8)). We both assume concavity of the utility function.
> 2. We require smoothness assumptions on the utility function (Assumption 3) whereas Mutti et al. only require Lipschitzness of the same function (Assumption 1 therein). This is because smoothness is important for deriving optimization guarantees as we make use of gradient information whereas Lipschitzness is enough for developing their statistical analysis.
> 3. Mutti et al. 2023 assume access to an optimal planner (as mentioned above), we do not need such a requirement as we provide optimization guarantees using our PG algorithm.
> 4. We need policy parametrization assumptions as previously discussed, Mutti et al. do not consider policy parametrization.
>
> We hope this discussion further clarifies the comparison to Mutti et al. 2023 and our contributions.

---

> > ### Author Response · Authors · 2024-11-25
> > **Response to second comment and questions**
> >
> > > Based on the content in the Appendix and my observations, the technical theoretical analysis appears to draw upon existing results. The statistical analysis is derived from [2], while the optimization analysis is based on [3], with minor adaptations to handle estimation error. However, this is not a critical issue if the authors address the concerns outlined in the Weakness 1, as this would help establish the paper’s distinct contributions more effectively.
> >
> > Yes, our analysis builds on prior work [2,3] as you mention it and as we acknowledge it in the main part and in the appendix. We point out here though that Barakat et al. 2023 does not provide global optimality guarantees. For this result, we rather rely on the hidden convexity analysis developed in Zhang et al. 2021 and adapt it for our purposes by carefully propagating the occupancy measure estimation error.
> >
> > **Question 1. (About global convergence and comparison to Barakat et al. 2023.)** Compared to [1] which does not provide global convergence guarantees, we require additional assumptions to obtain this stronger result. These are mainly assumptions 4  (concavity of the utility function, recall we are maximizing) and 5 (policy overparameterization assumption). We note that our function approximation regularity assumption (Assumption 1) is also different from [1] as we consider a different MLE estimation procedure. This assumption is also used for the first-order stationarity result though and it is not specific to the global optimality result. As mentioned above, our analysis builds on the hidden convexity analysis of Zhang et al. 2021 (where the same assumptions were made) and adapts it by carefully propagating the occupancy estimation error.
> >
> > **Question 2. (Comparison to Mutti et al. 2023).** Mutti et al. 2023 builds on a reduction to an existing online-learning result to prove their average regret guarantee (Theorem 5). Our last iterate global convergence result is different and relies on optimization analysis techniques combined with statistical guarantees. It comes at the price of the combination of the concavity assumption and Assumption 5. The latter assumption which was used in prior work (Zhang et al. 2021, Barakat et al. 2023) is quite restrictive as we mention it in the paper. It is crucial to be able to leverage the hidden convexity of the problem. However, it is about policy parametrization and there is no policy parameterization in Mutti et al. 2023. We also point out that the algorithm in Mutti et al. 2023 is statistically more efficient as can be shown from table 1 p. 14 if we insist on comparing them despite the fact that (a) we do not assume oracle access to convex MDP solvers (providing optimal policies), (b) our problem formulation is different, (c) Mutti et al. 2023 uses optimism which helps, (d) their regret bound still explicitly depends on the state action space sizes.
> >
> > **Question 3. (Theorem 2 and $\epsilon_{MLE}$).**
> > - In Theorem 2, we suppose that the error is uniformly bounded by $\epsilon_{MLE}$, which means that at each iteration $t$ of the algorithm, the estimation error for approximating the true occupancy measure induced by the policy $\theta_t$ is bounded by (the same) $\epsilon_{MLE}$. Therefore, $\epsilon_t$ is not dependent on T. This is because in the analysis we already propagated the error $\epsilon_{MLE}$ in the analysis. To be more precise, at each time step, the utility gap (difference between the utility induced by policy $\theta_t$ and the optimal utility) contracts by a factor (1-\eta) up to an error which is proportional to $\alpha \epsilon_{MLE}$ where $\alpha$ is the step size of the PG algorithm (see Eq. (47) p. 23) for the precise recursion at each time step). Then one unrolls the recursion over time up to the time horizon $T$ and the estimation error at each time step ($\alpha \epsilon_{MLE}$) is now amplified by the factor $1/\eta$ leading to an overall estimation error of ($\alpha \epsilon_{MLE}/\eta$) over the $T$ iterations. We obtain Theorem 2 this way.
> > - In Theorem 2, we suppose that the error is uniformly bounded by the same $\epsilon_{MLE}$ at each iteration, this can be guaranteed thanks to Proposition 1 because the upper-bound in that result is independent of the policy parameter $\theta$. The upper bound does only depend on the constant $B_{\omega}$ (Assumption 1 (i)) which is a uniform compactness constant bounding the set of parameters used in function approximation. This constant only appears inside the log term in the bound of Proposition 1. The overall intuition is that true occupancy measures are supposed to be realizable within a function approximation class which is covered in an appropriate way with a controlled size (see Prop. 2 p. 19, i.e Lemma 12 in Huang et al. 2023 and lemma 2 in the same page).

---

> ### Author Response · Authors · 2024-11-25
> **Response to question 4**
>
> **Question 4. (Dependence on the policy space size).**
> We use Theorem 2 and Proposition 1 combined to obtain Corollary 2, i.e. we use Proposition 1 to choose $\epsilon_{MLE}$ appropriately by setting the upper bound in Prop. 1 to be equal to $\epsilon_{MLE}$. This gives our conditions on the number of samples to draw $n$ for occupancy measure estimation and the dependence on the dimension $d$ of the function approximation class. As we suppose realizability (Assumption 1 (ii)), $\epsilon_{MLE}$ only depends on the size of the parameter space $\Omega$ used in the function approximation class rather than the size of the policy space (i.e. dimension of the policy space parameter $\theta$). This dependence is clear via (a) the dimension $d$ of the function approximation space which appears in the final sample complexity and  (b) an additional mild dependence captured by a constant inside the log terms in Proposition 1, namely the constant $B_{\omega}$, as the function approximation class ‘covers’ the set of occupancy measures induced by policies (see section D.2 for details).
>
> Thanks again for your review. Please let us know if you have any further comments or questions. We will be happy to address them.

---

> > ### Comment · Reviewer_Kpay · 2024-12-03
> >
> > I thank the authors for their detailed responses. I think it would be helpful if the authors could elaborate more on their contributions to the existing literature in the main text. I am considering raising my score to a 6.

---

> > > ### Author Response · Authors · 2024-12-04
> > > **Thank you**
> > >
> > > Thank you very much for your time, your useful feedback and your response to our rebuttal. We will surely add the points we have mentioned to our paper.

---

### Official Review · Reviewer_qbxS · 2024-11-10

**Soundness:** 3
**Presentation:** 3
**Contribution:** 1
**Rating:** 3
**Confidence:** 3

**Summary:**

The proposed method estimates state-action occupancy measures by MLE and uses them for policy gradient to maximize general utility.

**Strengths:**

1. MLE estimation of state-action occupancy measure is more sample-efficient than count-based algorithms and can be used in continuous state-action spaces

**Weaknesses:**

1. The proposed work learns state-action occupancy measures (Eq. 8). But it does not learn the occupancy measure defined in Eq. 1. In Eq 1, the occupancy measure is discounted by \gamma. However, Eq. 8 does not have a discount. And we don't know if the learned occupancy measure satisfies the backward Bellman flow equation which is mentioned in Section 3.1.


2. line 221: "Second and foremost, while prior work has used Monte Carlo estimates for
this quantity, such count-based estimates are not tractable beyond small tabular settings."

There are RL algorithms that focus on using stationary state-action distributions for learning policies that maximize expected returns with some constraint on the stationary distributions and these works do not use count-based estimates and works on continuous state-action space as well [1][2].

[1] Lee et al. "Coptidice: Offline constrained reinforcement learning via stationary distribution correction estimation." ICLR 2022.

[2] Mao et al. "Diffusion-DICE: In-Sample Diffusion Guidance for Offline Reinforcement Learning." NeurIPS 2024.

3. Question on the practicality of the proposed method: after each policy parameter update, you would need to gather data for the MLE of the occupancy measure and fit the occupancy measure which can be time-consuming.

**Questions:**

1. How does the proposed method relate to [1][2]? Additional experiments comparing the performance of these algorithms with the proposed method would also be helpful in understanding the difference between the methods. Since [1][2] are offline RL algorithms, maybe letting the target policy collect data periodically would make it possible to run them on on-policy manner and compare them to the proposed method.

[1] Lee et al. "Coptidice: Offline constrained reinforcement learning via stationary distribution correction estimation." ICLR 2022.
[2] Mao et al. "Diffusion-DICE: In-Sample Diffusion Guidance for Offline Reinforcement Learning." NeurIPS 2024.

2. Question on the practicality of the proposed method:  Additional experiments showing the time required until convergence of the target policy performance for the proposed method and the baselines would help clear this concern.

---

> ### Author Response · Authors · 2024-11-25
> **Rebuttal to reviewer qbxS**
>
> Thank you for your review. As a first comment regarding the presentation that you find ‘poor’, we highlight that all other reviewers are unanimous on the fact that the presentation is good, the paper well-written and easy to read, we refer the reviewer to their numerous individual positive comments. Please let us know if you have any meaningful constructive comment to further improve our work and we will be happy to take it into account. We provide a detailed point-by-point response to your comments in what follows:
>
> > The proposed work learns state-action occupancy measures (Eq. 8). But it does not learn the occupancy measure defined in Eq. 1. In Eq 1, the occupancy measure is discounted by \gamma. However, Eq. 8 does not have a discount.
>
> In contrast to what the reviewer is claiming, (Eq. 8) does allow us to learn the occupancy measure as we define it in (Eq. 1). Note that the samples used in (8) for the estimation are generated according to the **discounted** state action occupancy measure defined in (Eq. 1) which is indeed the discounted distribution. We invite the reviewer to Appendix D.1 (as we mentioned in l. 269 p. 5) to see our (standard) sampling procedure which allows us to generate such samples from the discounted distribution.
>
> > And we don't know if the learned occupancy measure satisfies the backward Bellman flow equation which is mentioned in Section 3.1.
>
> - There is no reason why the **approximated** occupancy measure should satisfy the Bellman flow equation in general. For instance, to draw a parallel with value function approximation, the approximated value function when considering linear function approximation might only satisfy a **projected** Bellman equation when using TD learning with linear function approximation. There are also a number of different approaches in the literature to incorporate function approximation: minimize the squared Bellman error, the projected Bellman error, the TD error, … each one leading to an algorithm (see e.g. Sutton, Szepesvári and Maei (2009) and the thesis of the last aforementioned author). Similarly here, we do not necessarily require the approximation to satisfy the Bellman flow equation (although we use some of its characteristics for learning it of course, see our sampling procedure used for learning our maximum likelihood estimator).
>
> - In addition, as we comment on in the paper, the occupancy measure does not satisfy a standard ‘forward’ Bellman equation which is immediately amenable to sampling as this is the case for standard value functions. To the best of our knowledge, there is no immediate way to address our online problem this way in a straightforward manner. This is one of the reasons why we follow a different approach.
>
> - Even in the works that the reviewer mentions (for standard RL or CMDPs), there is no guarantee that occupancy measure approximations (in the function approximation setting) will satisfy the Bellman flow equations. While these works do incorporate them into the formulation to derive their algorithm in the tabular setting, considering neural network parametrization does not guarantee to preserve any such favorable property of the original problem formulation which becomes another  problem formulation.
>
> > line 221: "Second and foremost, while prior work has used Monte Carlo estimates for this quantity, such count-based estimates are not tractable beyond small tabular settings."
> There are RL algorithms that focus on using stationary state-action distributions for learning policies that maximize expected returns with some constraint on the stationary distributions and these works do not use count-based estimates and works on continuous state-action space as well [1][2].
>
> As a first preliminary comment, **the references [1] and [2] that you mention do not address our general RLGU problem and do not provide any theoretical guarantee in terms of convergence nor sample complexity**. [1] deals with a CMDP problem whereas [2] focuses on the standard expected return RL problem using diffusion models in offline RL settings. In both these cases, estimating occupancy measures is not needed (at least in the online setting which is our focus). There are of course a number of works in the literature to address standard RL and CMDP problems using PG methods for instance (in the online setting) with theoretical guarantees, including for larger state action spaces. We highlight that **the main motivation of RLGU is to deal with objectives which are nonlinear function(al)s of the occupancy measure**: standard RL and CMDP problems are essentially linear programs in the occupancy measures (if we do not think about complex practical policy parametrization) and widely studied while also being particular cases of RLGU. In contrast, RLGU calls for estimating occupancy measures (at least if a PG approach is followed like in our work) if one wants to tackle the RLGU problem without ad hoc approaches tailored to special instances of RLGU.

---

> > ### Author Response · Authors · 2024-11-25
> > **Further discussion about the references and our goal**
> >
> > We maintain our claim in l. 221 (which relates to RLGU as it is clearly mentioned in l. 212 prefacing the paragraph) that we are not aware of any work addressing RLGU for larger scale state action spaces in both theory and practice: Prior work in RLGU has mostly focused on the tabular setting, using count-based estimates of occupancy measures.
> >
> >  **Further detailed discussion about [1] and [2] beyond the fact that they do not address our problem.** We provide a more detailed discussion regarding the approaches adopted in [1] and [2] while we invite the reader to keep in mind that they address different particular instances of our problem. We highlight **a number of additional differences with our work**:
> >
> > 1. **Offline vs online setting.** On a high level, all these works build on the so-called DICE line of works. We did cite one of the earlier works leading to such a line of research (Hallak and Mannor (2017), ‘Consistent Online-Off-Policy Evaluation’). To the best of our knowledge, one of the primary motivations of these works is to address the offline setting which is not immediately relevant to our online setting. Off-policiness is not a challenge we address here and we do not need to deal with it in the setting of our present work. We will add these works to our related work section though for completeness.
> >
> > 2. **Distribution corrections learning vs occupancy measure learning**. These works propose to learn the ratio between the stationary distribution of a target policy and that of the behavior policy (or policies) from which the data is issued. In contrast to this line of works, we propose to directly learn the occupancy measure. In our online setting, we do not see why such a ratio estimation can be needed nor why it could be useful. The reviewer argues that there are ‘RL algorithms that focus on using stationary distributions and these works do not use count-based estimates’ for different problems.
> >
> > 3. **Function approximators and hidden convexity structure of the problem.** Besides these differences, parameterizing the variables by representing them with function approximators actually breaks all the favorable properties of the original problem (as this is also mentioned in those works, see e.g. Lee et al.ICML 2021, OptiDICE). In particular the minmax theorem which is crucial to their problem formulation derivation does not hold anymore and the duality can now be arbitrary given the nonconvexity of the problem. We natively deal with policy parametrization from our starting problem formulation. In particular, besides the general nonconvex setting which is of most importance in practice and for which we provide first-order stationarity guarantees, we also exploit the hidden convexity of the problem building on prior work when the objective function is convex in the occupancy measure, i.e. we exploit this property even if the problem is nonconvex in the policy parameter.
> >
> > 4. **Regularization of the problem.** Their approach relies on regularization which changes the original problem. While this regularization might be justified in the offline setting, we note that (a) it is not necessarily meaningful/useful in ours, (b) it introduces a bias which cannot be readily annihilated and (c) it is crucial in their approach to obtain a closed form of the inner problem of their minmax problem formulation (which only makes sense in the tabular setting offering the nice convexity structure). We do not make use of such regularization in our work.
> >
> > 5. **Policy extraction.** Besides all these considerations, as the variables are now ratios in these works (distribution corrections), policy extraction becomes another important issue due to the need to estimate the unknown normalization constant for marginalizing the occupancy measure (especially for continuous state action spaces). We do not have to deal with such issues since we directly parametrize our policies and our optimization variables in RLGU are actually the policy parameters. Approaches to deal with this require to solve additional problems such as log-likelihood optimization problems (importance-weighted behavioral cloning) which are ironically variants instances of our RLGU problem formulation. [2] proposes some other approaches to avoid such requirements.

---

> > > ### Author Response · Authors · 2024-11-25
> > > **Conclusion about the discussion of the references and clarification of our goal**
> > >
> > > 6. **Linearity vs nonlinearity w.r.t. occupancy measures.** To conclude this discussion, we note that linearity of the objective (w.r.t. the occupancy measure) is important in the derivations of the transformed problem formulation of [1] (see e.g. Eqs. (6) and (7) in OptiDICE for instance). Such a linearity is not available in our setting and this renders their approach virtually unusable even if one wants to draw inspiration from such approaches. In addition to these considerations, there are a number of other approximations in OptiDICE that render any theoretical analysis hopeless. We will not expand on these in detail here but some of them (in addition to all the ones discussed above) include biasedness due to exchanging expectation and nonlinear functions and using upper bounds to the problem.
> > > We hope that this detailed discussion will help the reviewer better understand our contributions and the positioning of our work.
> > >
> > > **Our main goal.** The main goal of our work is to support our algorithm design with theoretical results including convergence and sample complexity guarantees under suitable assumptions. Most importantly, our problem formulation is general. To the best of our knowledge, solving RLGU (and not only its well-studied particular cases) in a computationally and statistically efficient manner is still a problem which is not carefully addressed in the literature. Our work contributes to addressing this question.
> > >
> > > **About the potential use of diffusion models.** That being said, we do not exclude that diffusion models might be of interest for our problem given that we are interested in solving a problem which is distributional in nature. We are not aware of any principled such approach and the work you shared does not answer this question.

---

> > > > ### Author Response · Authors · 2024-11-25
> > > > **Response to remaining questions**
> > > >
> > > > > Question on the practicality of the proposed method: after each policy parameter update, you would need to gather data for the MLE of the occupancy measure and fit the occupancy measure which can be time-consuming.
> > > >
> > > > **This is a point we discuss in the paper in the future work section.** Estimating the occupancy measure at each iteration is a current requirement of our algorithm. Note that even vanilla PG (say Reinforce) for standard RL does estimate the policy gradient at each policy parameter (from scratch) using Monte Carlo estimation. Therefore, estimating the occupancy measure estimation at each iteration is not unreasonable from this viewpoint. Nevertheless, we acknowledge that the occupancy measure estimation is of course a harder and more demanding problem to solve. In standard RL, actor-critic methods allow to use bootstrapping combined with Monte-Carlo to avoid estimating the full policy gradient at each policy update step to mitigate the sample complexity requirement you allude to. However, this relies on the fact that value functions satisfy the Bellman equation which is amenable to sampling. Recall that occupancy measures do not satisfy a similar (‘forward’) Bellman equation but rather ‘backward’ Bellman flows (conservation equation actually, see our remark). To lighten this estimation requirement, one possible strategy would be to consider regularized policy updates, we precisely described this possibility as future work in our submission (see l. 1337-1341, last page of the appendix).
> > > >
> > > > > How does the proposed method relate to [1][2]? Additional experiments comparing the performance of these algorithms with the proposed method would also be helpful in understanding the difference between the methods. Since [1][2] are offline RL algorithms, maybe letting the target policy collect data periodically would make it possible to run them on on-policy manner and compare them to the proposed method.
> > > >
> > > > **We do not intend to compare to existing algorithms for particular instances of our problem as [1][2] do not address our problem and do not provide any guarantee for their algorithm for their problem, let alone for ours.** We should mention that it is not even clear if any inspiration can be drawn from their approach to tackle our problem, let alone the fact that they focus on the offline setting. We address a more general problem in a unified way and our contribution is mainly at this level. Our goal is to address the RLGU problem with a simple and principled algorithm able to scale to larger state action spaces while also enjoying some convergence and sample complexity guarantees (under suitable assumptions of course which also have their own limitations as one might obviously expect).
> > > >
> > > > > Question on the practicality of the proposed method: Additional experiments showing the time required until convergence of the target policy performance for the proposed method and the baselines would help clear this concern.
> > > >
> > > > We will be happy to report the running time. We would like to highlight though that our point is that tabular count-based methods are clearly and fundamentally not scalable regardless of their running time on small scale problems.
> > > > We have implemented our method and tested it in several settings. To the best of our knowledge there is no existing practical (both computationally and statistically) algorithm for solving RLGU in general for large state action spaces that we can reasonably compare to. The work of Barakat et al. 2023 proposes an algorithm, however (a) they do not implement nor test their algorithm and (b) we argue in detail in our paper about the fundamental limitation in the way they even formulate the occupancy measure approximation problem. Please see also our response to reviewer MseJ regarding this point.

---

> ### Author Response · Authors · 2024-12-01
> **Response to the reviewer's comments**
>
> 2. **Our contributions** are mainly:  (a) proposing a simple model-free PG algorithm for RLGU replacing count-based occupancy measure estimators by scalable MLE which only use sampled trajectories, (b) showing that our approach is more suitable for the RLGU problem than the MSE approach proposed in prior work for the same problem (see our response to reviewer MseJ for clarifications) and (c) providing convergence and sample complexity guarantees for our algorithm which do not scale with the size of state action space. To the best of our knowledge, this is not covered by any of the works you mentioned (see our detailed discussion above) or any other work in the literature.
>
>
> 3. In general, function approximation errors in density estimation naturally depend on the class of probability distributions considered for the approximation. We mentioned Gaussian mixtures as an example in l. 254: ‘For continuous state spaces, practitioners can consider for instance Gaussian mixture models with means and covariance matrices encoded by trainable neural networks.’ For this specific example, one can try to tune the number of mixtures or even include this number in the learning parameters of the model. Of course, other parameterizations can be used. Note that the criticism you formulate can even be applied to policy parameterization. For continuous state action spaces, the most common policies used in RL in practice are Gaussians which are even unimodal and optimal policies need not be unimodal in general (note that Gaussian mixtures are at least multimodal).
>
> 4.
>
> **About the papers you mention.**
> - Ma et al. 2023 consider transition occupancy matching, this is similar to the standard imitation learning problem and they consider a **tabular** problem formulation. We are rather proposing to approximate the occupancy measure itself using function approximation in the more general RLGU setting. The Bellman flow constraint you mentioned is the same in all the papers you mentioned and we have also discussed it in our paper.
> - Yang et al., 2020 is about the same DICE approach that we have discussed in our response. The paper is focused on the policy evaluation problem in standard (expected return) RL (see their eq. (1)). They learn **occupancy ratios**. Similarly in Lee et al. 2021, for a similar problem, they learn occupancy ratios (a.k.a. stationary distribution corrections) as the title suggests. We have discussed this line of works and their limitations regarding our present motivation in our previous response.
>
> Learning occupancy measure **ratios (which is not our approach)** has been used **in some special instances of RLGU** in the works you mention as we discussed it. **This approach might not always be possible as the trick of considering the ratios and learning them only using offline data might not be possible** because the problem reformulation using these ratios might fail when considering objectives beyond standard RL or learning from demonstrations (involving ratios). Furthermore, **as we also mentioned, adopting such an approach also entails several additional difficulties to overcome** from which we highlight again two main ones:
> - One has to deal with an **intractable number of constraints** (even infinite uncountable in the continuous state action space setting) to incorporate Bellman flow constraints (supposing the Backward sampling issue can be handled), then using a Lagrangian approach introduces functional multipliers to handle this infinite dimensional problem which brings a number of technical difficulties. A possible approach would be to introduce function approximation at this stage but then an analysis would be required to see if any benefit can be concluded (how to introduce it? under which assumptions? …)
> - **Policy extraction from occupancy ratios** in large and continuous state action settings as we previously discussed.
>
>  **We propose a simple online approach to tackle our problem with theoretical guarantees.** Of course, we do not exclude that there might be other relevant approaches to tackle our problem and improve over our results.

---

> > ### Author Response · Authors · 2024-12-03
> > **Request to Reviewer**
> >
> > Dear Reviewer,
> >
> > We sincerely appreciate your time and effort to review our work. With the deadline for discussion ending in less than 20 hours, we want to make sure that our responses have addressed all of your concerns. Any additional insight would be very helpful to us. If all of your concerns have been addressed, we would request a reconsideration of your original score.
> >
> > Best,
> > Authors

---

### Comment · Area_Chair_e9pH · 2024-11-23
**From AC.**

Dear authors,

If possible, can you start the discussion about the paper by providing a rebuttal?

Thanks,
AC

---

> ### Author Response · Authors · 2024-11-24
>
> Dear AC,
>
> Thank you for your email, and I apologize for the delay.  We are going to post the rebuttal soon.
>
> Regards,
>
> Authors

---

### Meta-Review · Area_Chair_e9pH · 2024-12-19

**Metareview:**

The paper addresses the setting of general utility reinforcement learning i.e. the setting where utility is a general (not necessarily linear) functional of the state-action occupancy distribution.  The proposed algorithm works by estimating the occupancy using a variant of policy gradient.

The main strength of the work is that general utility reinforcement learning has many applications, while being underrepresented in RL literature.

However, reviewers mostly agree that the paper isn't ready. One particular point of concern is the novelty novelty relative to the work of Barakat et al. The authors do provide a comparison in Appendix A, but reviewers view the novelty as insufficient anyway. Another problem is the clarity of presentation.

For those reasons, I recommend rejection.

References:

Barakat et al.: Reinforcement Learning with General Utilities: Simpler Variance Reduction and Large State-Action Space

**Additional Comments On Reviewer Discussion:**

I was initially on the fence about this paper and considered championing it.

After following up with the reviewers, they convinced me that the novelty problem is real. Hence I recommend rejection.

---

### Decision · Program_Chairs · 2025-01-22

Reject